# Regulation of Rad52-dependent replication fork recovery through serine ADP-ribosylation of PolD3

Frederick Richards[1], Marta J. Llorca-Cardenosa[1], Jamie Langton[1], Sara C. Buch-Larsen [2], Noor F. Shamkhi[1], Abhishek Bharadwaj Sharma [1], Michael L. Nielsen [2] & Nicholas D. Lakin [1] ✉

Although Poly(ADP-ribose)-polymerases (PARPs) are key regulators of genome stability, how site-specific ADP-ribosylation regulates DNA repair is unclear. Here, we describe a novel role for PARP1 and PARP2 in regulating Rad52-dependent replication fork repair to maintain cell viability when homologous recombination is dysfunctional, suppress replication-associated DNA damage, and maintain genome stability. Mechanistically, Mre11 and ATM are required for induction of PARP activity in response to replication stress that in turn promotes break-induced replication (BIR) through assembly of Rad52 at stalled/damaged replication forks. Further, by mapping ADP-ribosylation sites induced upon replication stress, we identify that PolD3 is a target for PARP1/PARP2 and that its site-specific ADP-ribosylation is required for BIR activity, replication fork recovery and genome stability. Overall, these data identify a critical role for Mre11-dependent PARP activation and site-specific ADP-ribosylation in regulating BIR to maintain genome integrity during DNA synthesis.

Poly(ADP-ribose) polymerases (PARPs) are a cornerstone of the DNA damage response (DDR) that catalyse ADP-ribosylation (ADPr) by the addition of mono- or poly(ADP-ribose) moieties onto target proteins in response to genotoxic stress. The best characterised role of PARPs is in the regulation of DNA strand break repair[1,2]. PARP1 and PARP2 are activated upon binding DNA single strand breaks (SSBs) and ADP-ribosylate substrates to promote recruitment of XRCC1 to the break that in turn assembles chromatin remodelling and DNA repair factors to the damage site[2–4]. Importantly, disruption of this pathway with PARP inhibitors (PARPi) is toxic to cells with defects in homologous recombination (HR) due to their inability to perform replication-associated repair. PARPi treatment results in accumulation of unrepaired SSBs, or trapping of the PARP at DNA lesions, that when encountered by replication forks require HR-mediated repair[5–7]. Additionally, the BRCA pathway restrains PARPi-induced ssDNA gap formation behind replications forks and the levels of these structures correlate with PARPi sensitivity or resistance[8–11].

Intriguingly, PARPs also regulate various aspects of replication-associated repair. When replication forks encounter DNA lesions or blocks they are typically stabilised by reversal and annealing of the two nascent strands to form chicken foot-like structures[12–14]. Whilst fork remodelling proteins such as HLTF, ZRANB3 and SMARCAL1 promote the formation these structures, several HR proteins are also critical for their assembly and maintenance[15,16]. For example, Rad51 promotes replication fork reversal[14], whilst BRCA2 and Rad51 protect regressed replication forks from extensive degradation by Mre11/DNA2 in a manner that is mechanistically distinct from their role in DSB repair[17,18]. PARPs regulate these and other aspects of DNA replication including Okazaki fragment processing[19], recruitment of Mre11 to stalled/damaged replication forks[20–22], inhibition of RECQ1 helicase to maintain regressed forks[12], and stabilising HR factors at these structures[23].

Whilst the role of HR in replication fork recovery is well established, how other replication-associated repair mechanisms compensate for loss of this pathway and integrate into models of synthetic

[1]Department of Biochemistry, University of Oxford, South Parks Road, Oxford, UK. [2]Novo Nordisk Foundation Centre for Protein Research, Faculty of Health and Medical Sciences, University of Copenhagen, Blegdamsvej 3B, 2200 Copenhagen, Denmark. ✉e-mail: nicholas.lakin@bioch.ox.ac.uk

lethality between PARP1/2 and HR is unclear. For example, Rad52-dependent break-induced replication (BIR) promotes replication fork recovery[13,24–27] and consistent with this pathway functioning in parallel with HR, its disruption is toxic to HR-deficient cells[28,29]. This interaction may reflect a role analogous to yeast Rad52 by loading Rad51 at DSBs in *BRCA2*-defective cells[28,29]. However, the Rad51 binding domain of Rad52 is not required to maintain cell viability in *BRCA*-deficient backgrounds[30,31] and Rad52 depletion is synthetic lethal with other HR factors in *BRCA2*-proficient cells[29,32,33]. Together, these data suggest Rad52 regulates other important mechanisms beyond redundancy with the BRCA2-Rad51 axis to maintain cell survival in HR-defective cells.

Here we address this question by delineating a mechanism for synthetic lethality between Rad52 and HR that identifies an unanticipated role for PARPs in regulating replication fork recovery by BIR. We identify that PARPs and Rad52 function in the same pathway to maintain cell viability in the absence of HR, in addition to suppressing replication-associated DNA damage and genome instability. Mechanistically, we uncover that PARPs promote BIR-dependent replication fork recovery and that this regulation is achieved through Mre11-dependent PARP activation facilitating Rad52 assembly at stalled/damaged replication forks. Finally, by mapping ADPr events induced upon replication stress, we identify that site-specific ADPr of PolD3 is required for BIR, replication fork recovery and maintenance of genome integrity.

## Results

### PARP1 and Rad52 function in the same pathway to maintain cell viability in HR-defective cells

Our previous work identified that *PARP1* gene disruption is a major determinant of synthetic lethality with Rad51 inhibition[23]. Therefore, to identify novel PARP-dependent DNA repair mechanisms, we assessed whether other DDR genes are epistatic with *PARP1* gene deletion in terms of synthetic lethality with defective HR. Given *PARP1* gene disruption is toxic in HR-deficient cells, we generated reagents to conditionally disrupt BRCA1 expression in cells by inserting a Small Molecule-Assisted Shutoff (SMASh) degron[34] onto the endogenous *BRCA1* gene to allow BRCA1 depletion using asunaprevir (ASV; Fig. 1a and Supplementary Fig. 1a). As predicted, ASV depletes SMASh-BRCA1 in a time and concentration-dependent manner (Fig. 1a and Supplementary Fig. 1b). This results in an inability of cells to form Rad51 foci and sensitivity to DNA DSBs, indicating disruption of BRCA1-dependent DSB repair (Fig. 1b and Supplementary Fig. 1c). Exposure of *BRCA1*^SMASh cells to ASV also sensitises cells to a variety of PARPi (Fig. 1c and Supplementary Fig. 2). Consistent with *PARP1* gene disruption being synthetic lethal with Rad51 inhibition[23], knockout of the *PARP1* gene in *BRCA1*^SMASh cells confers sensitivity of cells to ASV-dependent depletion of BRCA1 (Fig. 1d). Under the experimental conditions employed here, *BRCA1*^SMASh*parp1*Δ cells are as sensitive to ASV as *BRCA1*^SMASh cells exposed to PARPi, indicating this is a robust approach to assess synthetic lethal interactions with HR-deficiency (Fig. 1d).

Having established a system to assess the synthetic lethal interaction between HR and *PARP1* gene disruption, we sought to determine how this interaction integrates with other DNA repair mechanisms. For example, whilst Rad52 is required for cell viability in the absence of HR, the nature of this relationship is unclear. Therefore, we exploited this system to test whether Rad52 and PARP1 function in the same pathway to maintain cell viability when HR is dysfunctional. Consistent with a synthetic lethal interaction between Rad52 and HR, siRNA depletion of Rad52 sensitises *BRCA1*^SMASh cells to ASV (Fig. 1e and Supplementary Fig. 3a). Strikingly, depletion of Rad52 does not further sensitise *BRCA1*^SMASh*parp1*Δ cells to ASV, indicating Rad52 and PARP1 are epistatic and function in the same pathway to maintain cell viability when HR is dysfunctional (Fig. 1e). Collectively, these data identify that

*PARP1* gene disruption is synthetic lethal with defective HR through a mechanism that is dependent on Rad52.

### PARP1 and PARP2 regulate BIR to maintain genome stability in response to replication stress

Next, we wished to define the PARP1-dependent pathway that Rad52 regulates to maintain cell viability when HR is dysfunctional. Whilst yeast Rad52 promotes HR by loading Rad51 at sites of DSBs[27], in vertebrates this function is performed by BRCA2[35]. Instead, the principal role of vertebrate Rad52 is to regulate BIR, a homology-directed repair mechanism critical for a variety of processes including alternative lengthening of telomeres (ALT)[36,37], mitotic DNA synthesis (MiDAS)[38,39] and replication fork recovery[13,24–27]. Therefore, we initially tested whether PARP1/PARP2 regulate BIR by exploiting a previously established GFP reporter assay to quantify BIR-dependent repair at an I-Sce1-induced DSB[26]. In this assay the I-Sce-1 site separates two halves of a GFP reporter cassette and induction of DSBs through expression of I-Sce-1 allows reconstitution of GFP through recombination-based repair mechanisms. In this specific assay, repair by synthesis-dependent strand annealing (SDSA) or single-strand annealing (SSA) is prevented through a lack of sequence homology both sides of the I-Sce-1 break and the opposite orientations of the GFP cassettes. Therefore, GFP-reconstitution occurs through BIR-dependent DNA repair. Consistent with previous reports[26], sequencing of PCR products using primers that amplify the GFP cassette confirms reconstitution of GFP, indicating accurate recombination-based repair. Moreover, I-Sce-1 induced GFP expression is reduced upon Rad52 inhibition, or depletion of PolD3, confirming this event is dependent on BIR (Fig. 2c and see below). Consistent with a requirement for PARP1/PARP2 in BIR, whilst I-Sce1 expression in U2OS results in a significant increase in GFP-positive cells, this is compromised in *parp1/2*Δ U2OS cells (Fig. 2a and Supplementary Fig. 4). Additionally, the PARPi veliparib and PJ-34 reduce I-Sce1-induced GFP reconstitution in two independent U2OS clones containing the BIR-GFP reporter (Fig. 2b, c). However, when PARPi are combined with Rad52i there is no additional reduction in GFP induction (Fig. 2c), indicating that Rad52 and PARP1/2 function in the same pathway to restore GFP expression through BIR.

Next, we considered whether PARP1/2 regulate Rad52 in the context of replication stress. Rad52 regulates two principal mechanisms in this context; BIR-dependent replication fork recovery[24–27] and MiDAS, a process that completes replication and/or repair of atypical DNA structures during mitosis[38,39]. We observe little impact of *PARP1/2* gene disruption on replication stress-induced MiDAS (Supplementary Fig. 5), indicating PARP1/2-dependent ADPr is not required for BIR in this context. Consistent with its role in replication fork recovery, siRNA depletion of Rad52 elevates levels of HU-induced DSBs (Fig. 2d). However, it does not induce a further increase in DSBs in *parp1/2*Δ cells, indicating PARP1/2 and Rad52 suppress HU-induced DNA breaks through a shared mechanism (Fig. 2d). Additionally, depletion of Rad52 does not further increase the levels of anaphase bridge or micronuclei formation evident in *parp1/2*Δ cells, indicating they function in the same pathway to maintain genome stability in response to replication stress (Fig. 2e, f). Together, these data identify that whilst PARP1/2 are not required for MiDAS, they regulate a Rad52-dependent mechanism to supress DNA damage and genome instability in response to replication stress.

### Mre11-dependent activation of PARP1/PARP2 is required for Rad52-dependent replication-associated DNA repair

Having established that PARP1/2 and Rad52 function in the same pathway to suppress replication-associated DNA damage, we next investigated the mechanisms of this regulation. Robust induction of ADPr is apparent in cells following exposure to HU that is primarily PARP1-dependent, with residual ADPr in *parp1*Δ cells being mediated by PARP2 (Fig. 3a). Given PARP1 responds to DNA strand breaks,

 

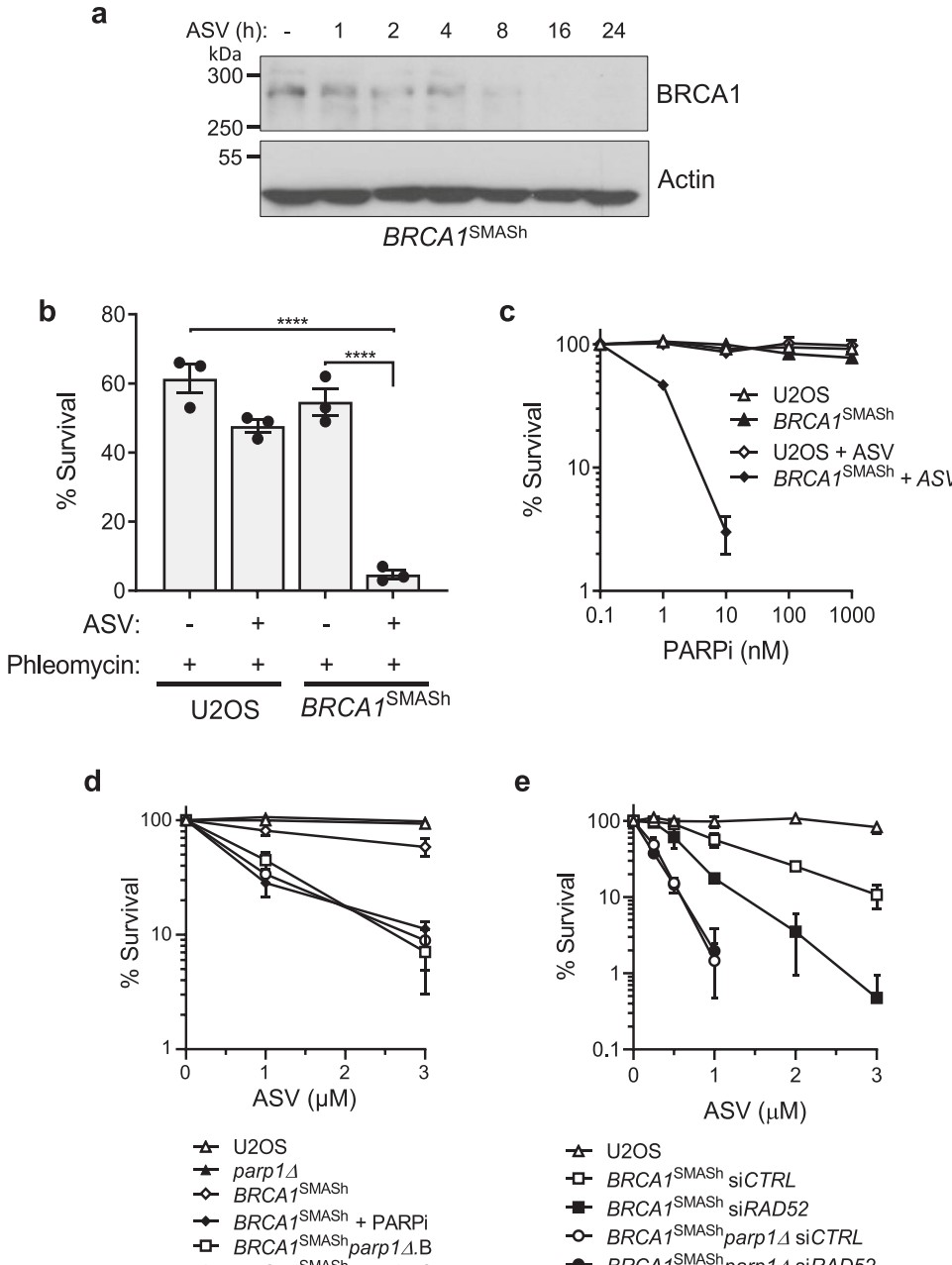

**Fig. 1 | PARP1 and Rad52 act within the same pathway to maintain viability in HR-defective cells. a** Inducible depletion of BRCA1-SMASh detected in whole cell extracts from *BRCA1*SMASh cells treated with 3 μM ASV for the indicated times. **b** ASV-dependent sensitivity of parental U2OS and *BRCA1*SMASh cells to DSBs induced by phleomycin following depletion of BRCA1-SMASh using 3 μM ASV as indicated. Survival is expressed as a % of untreated controls. Statistical analysis was performed by one-way ANOVA with Tukey's test applied post-hoc. Statistical significance was calculated to a confidence of 95%, $p < 0.05$ (*), 99%, $p < 0.01$ (**) or 99.9%, $p < 0.001$ (***). **c** Sensitivity of *BRCA1*SMASh cells following continuous exposure to increasing concentrations of olaparib (PARPi) as assessed by clonogenic survival assay

following depletion of BRCA1-SMASh using 3 μM ASV. **d** Clonogenic survival assay assessing survival of two independent *BRCA1*SMASh*parp1Δ* clones (*BRCA1*SMASh*parp1Δ*.B and *BRCA1*SMASh*parp1Δ*.C) continuously exposed to increasing concentrations of ASV, as indicated. Where indicated, *BRCA1*SMASh cells were additionally exposed to 500 nM olaparib (*BRCA1*SMASh + PARPi). **e** Clonogenic survival assay assessing effects of *RAD52* depletion on U2OS, *parp1Δ*, *BRCA1*SMASh and *BRCA1*SMASh*parp1Δ* cells exposed to increasing concentrations of ASV as indicated. Error bars represent the S.E.M of three independent experiments. Source data are provided as a Source Data file.

we hypothesized that processing of replication forks by nucleases such as Mre11, DNA2 and Slx4/Mus81[40,41], might be required to generate structures that activate PARP1/2. Depletion of Slx4, Mus81 or DNA2 does not affect induction of ADPr in response to HU (Fig. 3b, c and Supplementary Figs. 3c–e, 6 and 7a). In contrast, depletion of Mre11, or exposure of cells to the Mre11 inhibitor Mirin, reduces HU-induced ADPr to levels observed in untreated cells (Fig. 3c, d and Supplementary Figs. 3f and 7a). Given Mre11 is

required for ATM activation and that this has been implicated in processing replication-associated DNA breaks[42–44], we also considered whether ATM activity is required for ADPr in response to replication stress. Consistent with this, we observe that ATM inhibitors (ATMi) similarly reduce the levels of HU-induced ADPr (Supplementary Fig. 7b). Therefore, Mre11 and ATM-dependent processing of stalled and/or damaged replication forks is required for PARP1/2-dependent ADPr in response to replication stress.

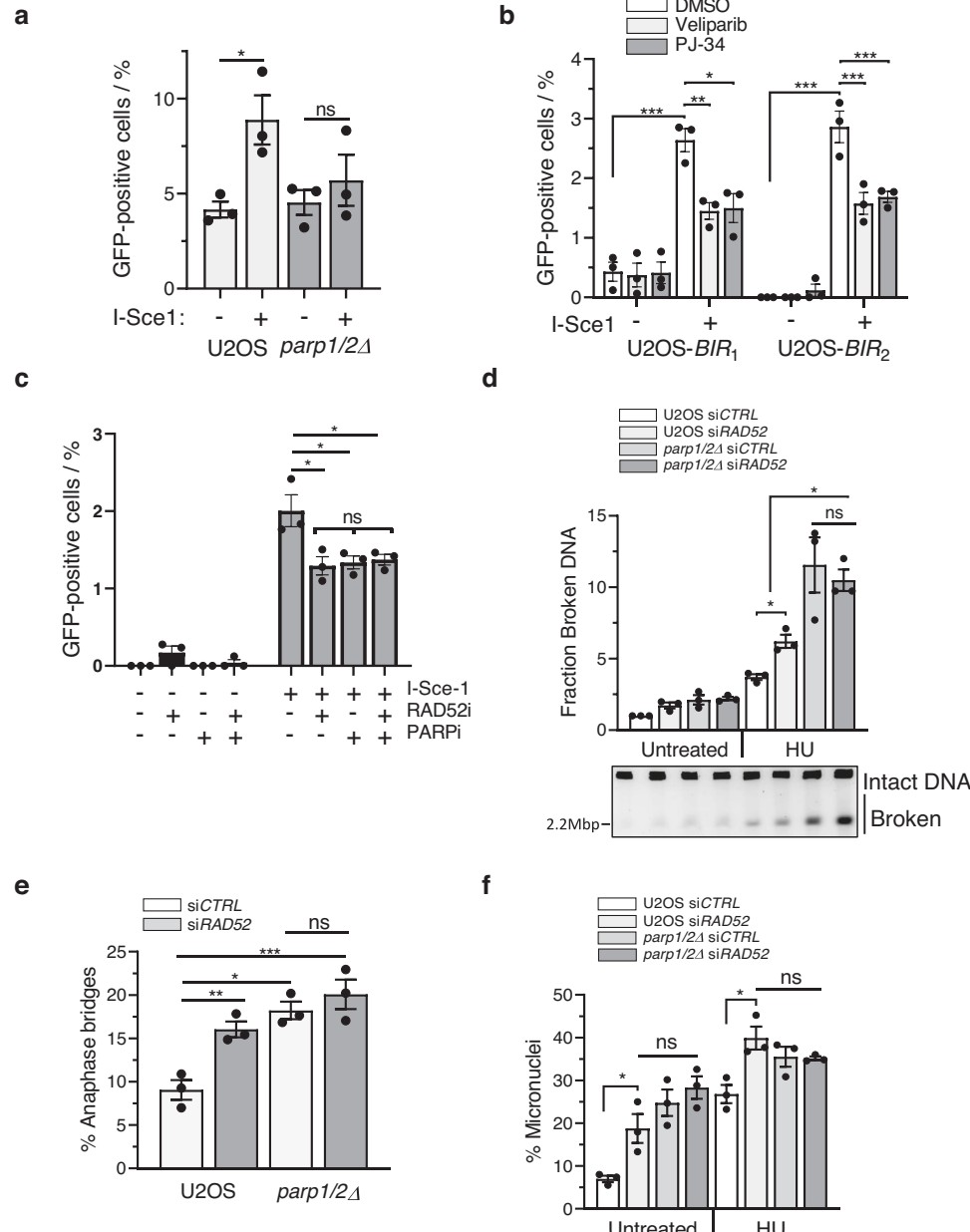

**Fig. 2 | PARP1 and PARP2 are required for BIR activity and the regulation of Rad52-dependent replication stress resolution. a–c** Quantification of GFP-positive U2OS and *parp1/2Δ* cells (**a**) or two independent U2OS clones (U2OS-*BIR₁* and U2OS-*BIR₂*) stably integrating the *BIR GFP* reporter and treated with PARPi (**b**) or Rad52i (**c**). **a** GFP fluorescence was analysed by flow cytometry 72 h after transient *I-SCE1* transfection. **b**. 8 h after initial *I-SCE1* transfection, U2OS-*BIR₁* and U2OS-*BIR₂* cells were treated with PARP inhibitors PJ-34 (20 μM), veliparib (Velip, 10 μM) or DMSO for 48 h before quantification of GFP expression by live fluorescence imaging. **c**. 8 h after initial *I-SCE1* transfection, U2OS-BIR₁ cells were treated with 20 μM PJ-34 (PARPi) and 5 μM 6-OH-DOPA (RAD52i), alone or in combination. After 48 h, GFP-expression was quantified by live cell imaging. **d–f** U2OS and *parp1/2Δ* cells were targeted for siRNA-mediated *RAD52* depletion and treated with 2 mM

HU for 24 h, as indicated. **d** DNA breaks were resolved from intact structures by pulse-field gel electrophoresis. The fraction of total DNA containing breaks was quantified and made relative to untreated U2OS cells. Inset below: a representative agarose gel image showing intact and broken DNA species stained with ethidium bromide. **e** DAPI-positive bridge formation in unperturbed anaphase cells. At least 200 cells were scored for each experiment. **f** Micronuclei formation scored 48 h after HU recovery. At least 400 cells were analysed per experiment. Error bars represent the S.E.M of three independent experiments. Statistical analysis was performed by one-way ANOVA with Tukey's test applied post-hoc. Statistical significance was calculated to a confidence of 95%, $p < 0.05$ (*), 99%, $p < 0.01$ (**) or 99.9%, $p < 0.001$ (***). Otherwise, analyses were classified as not significant (ns). Source data are provided as a Source Data file.

Following replication fork processing by Mre11, Rad52 assembles at the fork to coordinate Mus81-dependent cleavage and initiate BIR by facilitating strand invasion and PolD3-dependent DNA replication[24–27,45]. To delineate how ADPr regulates BIR further, we assessed whether any of these events require Mre11-dependent PARP1/PARP2 activation. Strikingly, *parp1/2Δ* U2OS cells, or *parp1Δ* RPE-1 cells depleted for PARP2, are unable to form Rad52 nuclear foci in response to replication

stress (Fig. 3e and Supplementary Figs. 3i and 8). HU-dependent accumulation of Rad52 in chromatin is similarly reduced in *parp1/2Δ* cells (Fig. 3f), further supporting the requirement for PARP1/2-dependent ADPr in promoting the recruitment and/or retention of Rad52 at stalled/damaged replication forks. Given Mre11 is required for ADPr in response to HU, a prediction of this model would be that Mre11 is also required to assemble Rad52 at sites of replication stress to promote BIR and as such

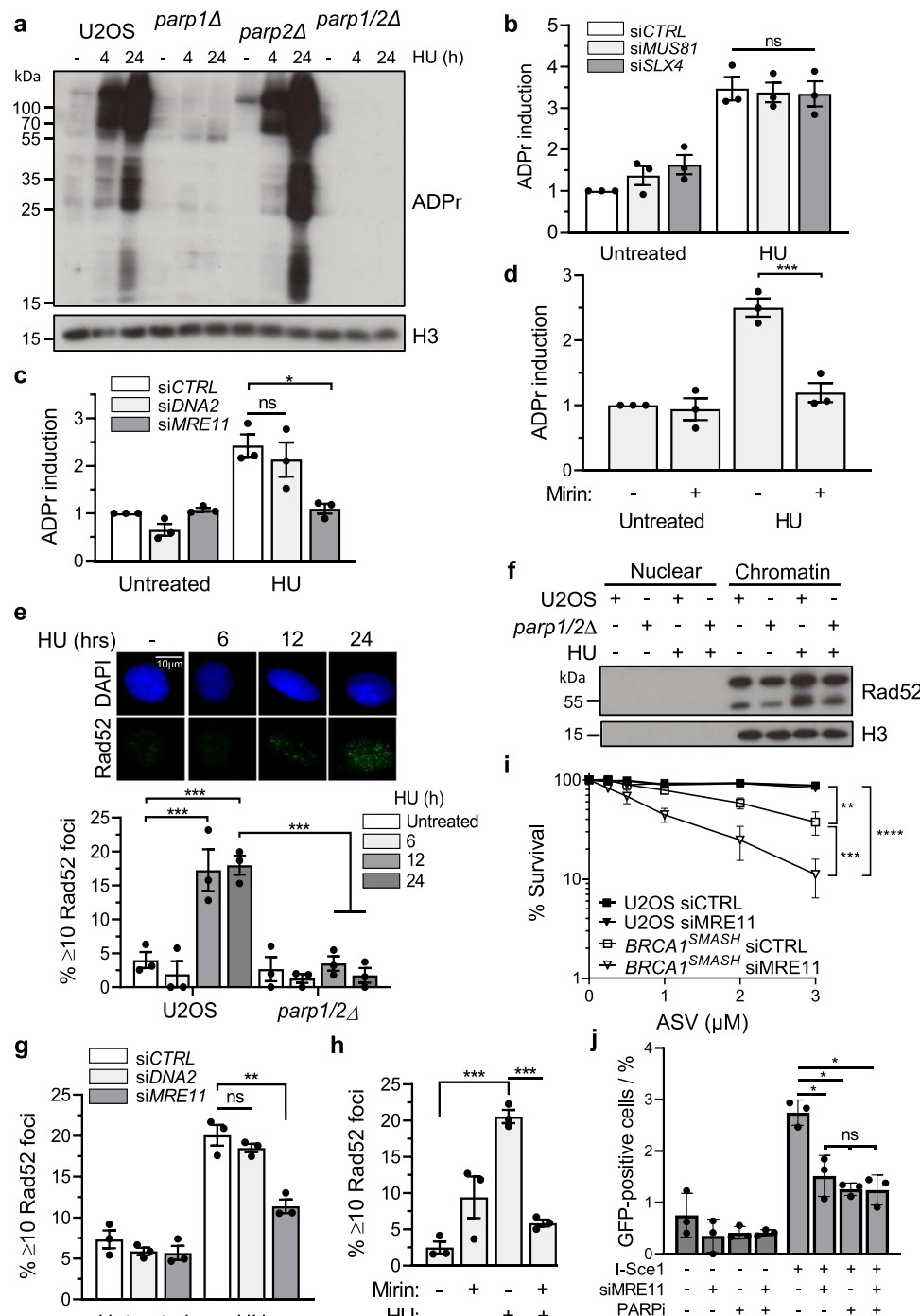

**Fig. 3 | Mre11-dependent activation of PARPs promotes assembly of Rad52 around HU-induced stalled forks. a** Assessment of the ADPr response in U2OS, *parp1Δ*, *parp2Δ*, and *parp1/2Δ* cells following exposure to 2 mM HU for the indicated times. Whole cell extracts were analysed by SDS-PAGE and total ADP-ribose levels were detected by immunoblot using a PAN-ADP-ribose binding reagent (ADPr). Images are representative of 3 independent experiments. **b–d** Nuclear ADPr induced following 24 h HU treatment was detected by immunofluorescence (A.U.) in U2OS cells depleted of Slx4 and Mus81 (**b**), DNA2 and Mre11 (**c**), or in the presence of the Mre11 inhibitor mirin (50 μM) (**d**). At least 1000 cells were analysed in each experiment. ADPr induction was calculated relative to levels in untreated control (siCTRL) cells. **e** Frequency of RAD52-positive nuclei actively incorporating EdU prior to 2 mM HU treatment for the indicated times. Representative images showing RAD52 foci are illustrated. An average of 100 cells were scored for each experiment. **f** Biochemical fractionation of U2OS and *parp1/2Δ* cells treated with 2 mM HU for 24 h. Nuclear soluble and chromatin fractions were analysed by Western blotting using the indicated antibodies. **g**, **h** Quantification

of RAD52-positive nuclei induced by 24 h HU exposure in U2OS cells depleted of DNA2 and Mre11 (**g**), or in the presence mirin (**h**). Each experiment scored at least 200 cells. **i** Clonogenic survival assay assessing sensitivity of *BRCA1*[SMASh] cells to MRE11 siRNA depletion following depletion of BRCA1-SMASh using concentrations of ASV as indicated. **j** Quantification of BIR activity at an I-SCE1-induced break following MRE11 knockdown and/or PARP inhibition. Following siRNA-mediated depletion of *MRE11*, U2OS-BIR₁ cells were transfected with *I-SCE1* for 8 h and then PARPi (20 μm PJ34) were added for a further 48 h. GFP expression was quantified by live cell imaging after subsequent 48 h PARPi exposure (20 uM PJ-34). Source data are provided as a Source Data file. In all instances, error bars represent the S.E.M of three independent experiments. Statistical analysis was performed by one-way ANOVA with Tukey's test applied post-hoc, with the exception of (i) which was performed using two-way ANOVA. Statistical significance was calculated to a confidence of 95%, $p < 0.05$ (*), 99%, $p < 0.01$ (**) or 99.9%, $p < 0.001$ (***). Otherwise, analyses were classified as not significant (ns). Source data are provided as a Source Data file.

would be synthetic lethal with BRCA1 depletion. Consistent with this, HU-induced Rad52 foci are compromised either upon depletion of Mre11, or exposure of cells to Mirin (Fig. 3g, h and Supplementary Fig. 9) and siRNA of Mre11 renders $BRCA1^{SMASh}$ cells sensitive to ASV (Fig. 3i). Moreover, we observe that depletion of Mre11 compromises BIR (Fig. 3j). Importantly, whilst PARPi similarly result in reduced BIR activity, this is not exacerbated when they are administered in combination with Mre11 depletion, indicating they regulate a shared mechanism to promote BIR (Fig. 3j). In summary, these data indicate Mre11-dependent activation of PARP1/PARP regulates BIR-dependent replication for recovery by promoting assembly of Rad52 at stalled/damaged replication forks.

## Mapping site-specific ADPr events induced upon replication stress

Although the role of PARP1 and PARP2 in maintaining genome integrity is well established, how ADPr regulates the repair process is ill-defined. Recent advances in mass spectrometry (MS) have begun to define the ADP-ribosylome[46–51] and a significant advance in our understanding was the identification that serine is the major ADP-ribose acceptor in response to DNA damage[48,52,53]. Nevertheless, which proteins are ADP-ribosylated in response to genotoxic stress and how this regulates the repair process is poorly understood.

To address this question in the context of replication stress, we exploited MS to identify ADPr sites induced in response to replication fork stalling and/or collapse. U2OS cells were left untreated, or exposed to HU, and ADP-ribosylated peptides enriched from cell lysates by affinity purification using the Af1521 ADP-ribose-binding macrodomain[54]. Purified peptides were subsequently analysed for site-specific ADPr events using quantitative MS (Fig. 4a)[50]. This revealed total ADPr intensity increases with HU exposure, particularly after prolonged replication fork stalling (Fig. 4b). Overall, we identified 199 ADPr sites (localization probability >0.75) residing on 118 proteins (Supplementary Data 1). The majority of these sites (~70%) are specifically induced by HU, with an additional 59 being identified in both conditions (Supplementary Fig. 10a). ADPr sites increase over time (Fig. 4c), and consistent with previous reports[48,52,53,55], serine represented the major ADPr acceptor residue in response to HU, with 84% of sites localised with confidence across all conditions (Fig. 4d). ADPr sites have a tendency to be placed within basic environments (Supplementary Fig. 10b). In the majority of cases, basic residues (R/K) were placed N-terminal to Ser-ADPr, with 'KS' motifs being the most abundant (Supplementary Fig. 10c, d).

Gene ontology (GO) analysis reveals a strong preference for ADPr of proteins within the nuclear compartment (89%; Supplementary Fig. 10e). Similar to observations in response to oxidative stress[52], proteins targeted at non-serine residues were more likely to reside in the cytosol (14%) compared to nuclear-biased serine sites (9%). Overall, enriched terms include processes relating to RNA processing, chromatin organisation, DNA replication and DNA repair (Fig. 4e). Whilst proteins involved in general chromatin binding/remodelling and RNA regulation were apparent across all experimental conditions, their fold enrichment generally decreased in response to replication stress (Fig. 4f). In contrast, DNA repair and replication terms are induced by HU, reflecting the replication-associated nature of DNA damage inflicted by this genotoxin. Intriguingly, specific DSB repair pathways, and NHEJ in particular, increase at later times following HU exposure (Fig. 4f), perhaps reflecting initial fork remodelling events before eventual collapse and engagement of repair mechanisms such as HR and/or NHEJ. Overall, these data highlight a critical requirement for ADPr in regulating multiple aspects of the replication stress response, including temporal target modification to influence distinct repair outcomes.

## Site-specific ADPr of PolD3 is required for replication fork recovery by BIR

Next, we considered specific ADPr sites induced upon replication stress and how this might contribute towards replication fork recovery. Of particular interest to this study, we identify the BIR protein PolD3 is ADP-ribosylated at S422 in response to replication stress. Importantly, ADPr of this site is either absent[50,52,53,56] or at low abundance[49] in studies that employ $H_2O_2$ to induce predominantly DNA strand breaks, underscoring the potential role of S422 in maintaining genome integrity during DNA replication. Given our findings that PARP1/2 promote genome stability through regulating Rad52-dependent replication fork repair, we therefore assessed the requirement for PolD3 S422 in these processes.

Independent affinity purification of ADP-ribosylated proteins confirms PolD3 ADPr (Supplementary Fig. 10f). To specifically assess the ADPr of PolD3 at S422 in response to replication stress we expressed siRNA-resistant wild-type (PolD3$^{WT}$) or S422A mutant PolD3 (PolD3$^{S422A}$) in cells (Supplementary Fig. 11a). Following replication fork stalling induced by HU, PolD3 Ser-ADPr was assessed by immunoprecipitation of recombinant protein from cell extracts and western blotting with an antibody that specifically recognises mono-ADPr[57]. Mono-ADPr of wild-type PolD3 is evident in cells either in the absence or presence of HU (Fig. 5a). However, whilst mono-ADPr of PolD3$^{S422A}$ is evident in untreated cells this is lost in response to HU (Fig. 5a), indicating that whilst S422 is not a major ADPr acceptor in unstressed cells, it plays a significant role in determining the mono-ADPr status of PolD3 in response to HU.

To probe the functional significance of this observation, we initially assessed the requirement for Ser-ADPr to assemble PolD3 into chromatin following replication stress. Consistent with a role in detection and/or repair of stalled replication forks, we observe enrichment of PolD3 in chromatin following exposure of cells to HU with similar kinetics to that observed for Rad52 foci formation (Fig. 5b–e and Supplementary Fig. 12). Importantly, this event is sensitive to PARPi (Fig. 5b) and $HPF1$ gene knockout (Fig. 5c), a factor that confers Ser-ADPr activity on PARP1/2[53]. Conversely, elevated levels of PolD3 chromatin association are apparent in $ARH3$ knockout cells (Fig. 5d and Supplementary Fig. 13), a gene that removes Ser-ADPr from target proteins[58]. Furthermore, we observe that HU-induced assembly of PolD3$^{S422A}$ in chromatin is reduced relative to PolD3$^{WT}$ (Fig. 5e). Together, these data indicate PARP catalytic activity, HPF1-dependent Ser-ADPr and PolD3 S422 are required to assemble PolD3 at stalled and/or damaged replication forks.

Next, we assessed the requirement for Ser-ADPr and PolD3-S422 in replication fork recovery by BIR. We observe a reduced efficiency of BIR at an I-Sce1 induced DSB in $HPF1$ depleted cells, indicating a requirement for Ser-ADPr in BIR (Fig. 6a; Supplementary Fig. 3g). As described previously[26], depletion of PolD3 results in an inability of cells to repair an I-Sce1 induced DSB by BIR (Fig. 6b and Supplementary Fig. 3h). Strikingly, whilst this phenotype is rescued by expression of PolD3$^{WT}$, PolD3$^{S422A}$ is unable to do so, indicating that S422 is required for BIR (Fig. 6b). DNA fibre analysis revealed that depletion of endogenous PolD3 had little impact on replication origin firing in the absence or presence of HU, and that this remains unaffected upon expression of PolD3$^{WT}$ or PolD3$^{S422A}$ (Supplementary Fig. 14). However, PolD3 depletion does result in increased termination events and/or replication fork stalling, and consistent with a role for PolD3 in replication fork recovery by BIR, replication fork restart is compromised following removal of HU (Fig. 6c and Supplementary Fig. 14). Whilst expression of PolD3$^{WT}$ rescues these phenotypes, PolD3$^{S422A}$ is unable to do so, indicating a requirement for S422 in supressing fork stalling events and promoting recovery in response to replication stress. Consistent with these observations, PolD3$^{S422A}$ is similarly unable to rescue the HU-sensitivity or micronuclei induction upon replication stress of PolD3-depleted cells (Fig. 6d, e), further confirming the

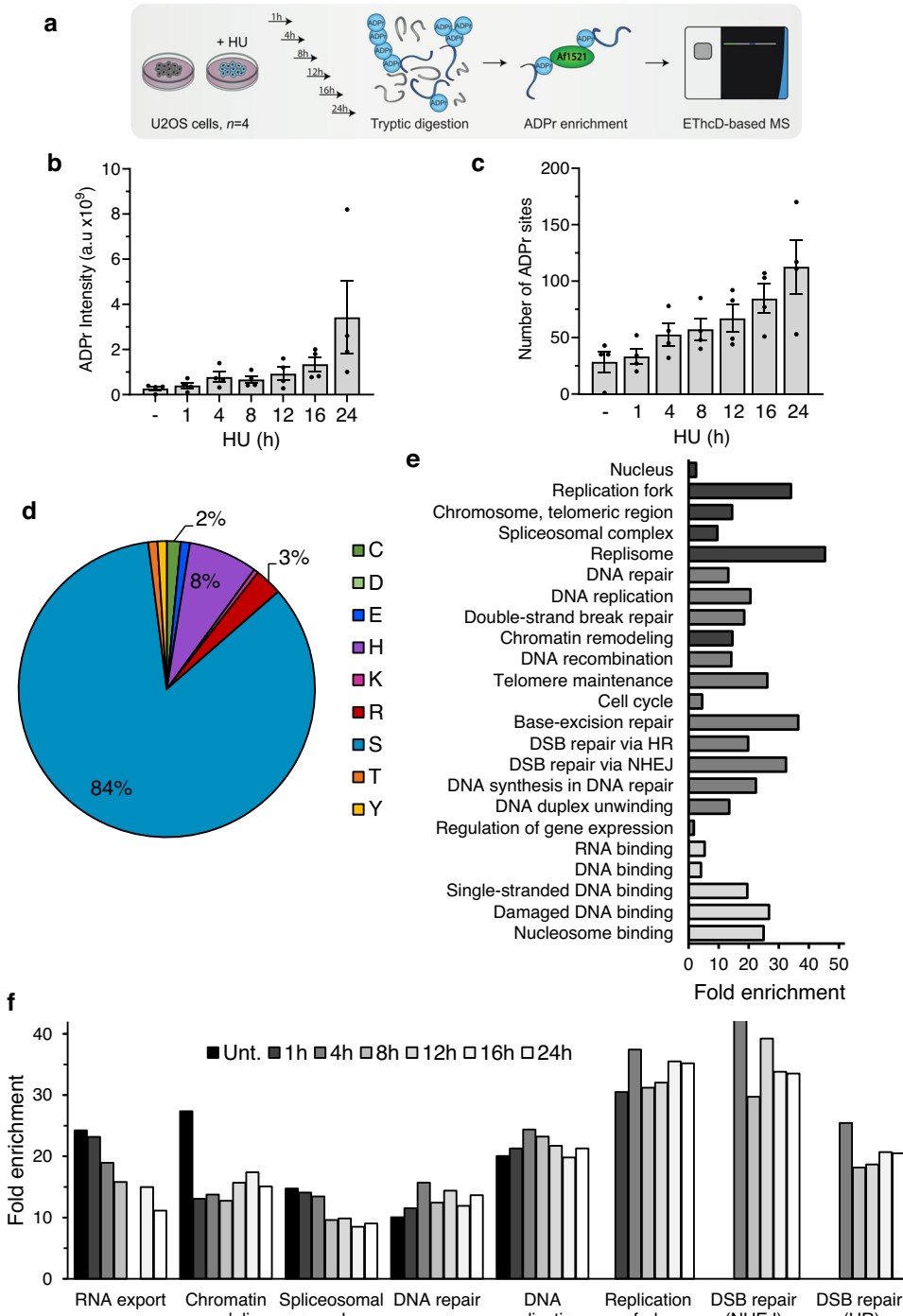

**Fig. 4 | System-wide identification of proteins targeted for ADPr in response to HU. a** Overview of experimental design to map ADPr sites. **b** Total intensity of ADPr signal detected by mass spectrometry after 2 mM HU treatments for the indicated time. **c** Number of ADPr sites confidently localised at different time points after HU exposure. **d** Pie chart showing amino acid distribution of all identified ADPr sites. **e** Gene ontology term enrichment for proteins targeted for ADPr after HU compared to the total genome. Significant terms were selected with FDR $p < 0.005$. **f** HU-induced profiles of GO term enrichment associated with RNA processing, chromatin organisation and replication-associated repair. In **b** and **c**, data are presented as mean values and error bars represent +/- SEM of four independent experiments. Source data are provided as a Source Data file.

importance of this site in maintaining genome integrity. Over-expression of PolD3[S422A] also results in defective BIR, replication fork recovery and genome instability in siRNA control cells that express endogenous PolD3. Given PolD3[S422A] interacts with other components of the PolD3 complex (PolD1 and PolD3; Supplementary Fig. 11b), we believe this is due to a dominant negative affect of the PolD3[S422A] mutant competing with endogenous PolD3, further underscoring the importance of S422A in replication for recovery mechanisms. In summary, these data indicate that Ser-ADPr and PolD3 S422 are required for BIR-dependent replication fork recovery to maintain cell viability and genome stability in response to replication stress.

## Discussion

Our previous work identified that PARP1 gene disruption is a major determinant of synthetic lethality with HR dysfunction[23]. Here we extend these studies by developing a strategy to conditionally deplete

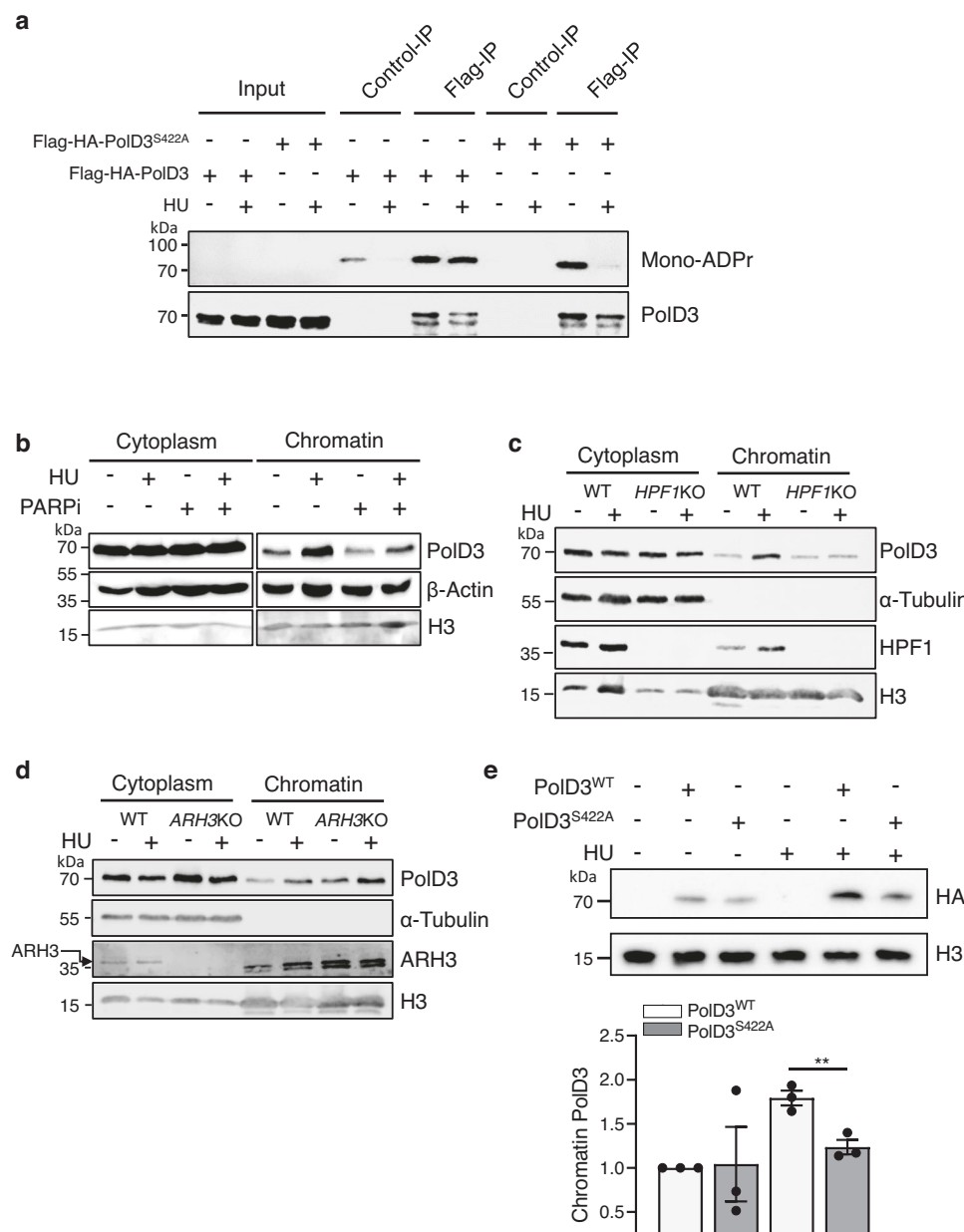

**Fig. 5 | PolD3 is ADP-ribosylated in response to HU and is required for its assembly into chromatin following replication stress. a** Mono-ADP-ribosylation of PolD3 in response to HU. U2OS cells expressing wild-type FLAG-HA-PolD3 or FLAG-HA-PolD3$^{S422A}$ were left untreated or exposed to 2 mM HU for 24 as indicated. Following extract preparation and immunoprecipitation with beads alone (Control-IP) or FLAG-beads (FLAG-IP) western blotting was performed with the indicated antibodies. Data are representative of three independent experiments. **b** Assessment of PolD3 Chromatin-bound levels in U2OS WT cells following exposure to 2 mM HU for 24 h either in the absence or presence of 20 μM PJ-34, as indicated. Cytoplasm and Chromatin biochemical fractionated extracts were analysed by SDS-PAGE and western blotting using the indicated antibodies. Data are representative of 3 independent experiments. **c**, **d** Assessment of PolD3 Chromatin-bound levels in HPF1 (c) and ARH3 (**d**) knockout (KO) U2OS cells following

exposure to 2 mM HU for 24 h. Cytoplasm and Chromatin biochemical fractionated extracts were analysed by SDS-PAGE and western blotting using the indicated antibodies. Data are representative of 2 independent experiments. **e** Chromatin enrichment of stably expressed FLAG-HA-tagged PolD3$^{WT}$, and mutant PolD3$^{S422A}$ in U2OS cells treated with 2 mM HU for 24 h. Where no recombinant PolD3 is indicated, cells were transfected with empty vector. Right panel: Quantification of FLAG-HA-PolD3 chromatin enrichment relative to PolD3$^{WT}$ levels in untreated cells. Error bars represent the S.E.M of three independent experiments. Statistical analysis performed using a one-way ANOVA with Tukey's test applied post-hoc. Statistical significance was calculated to a confidence of 95%, $p < 0.05$ (*), 99%, $p < 0.01$ (**) or 99.9%, $p < 0.001$ (***). Otherwise, analyses were classified as not significant (ns). Source data are provided as a Source Data file.

BRCA1 in combination with *PARP1* gene disruption, offering the possibility to identify novel genetic interactions that function alongside PARP1 to supress the toxicity of HR defects. Intriguingly, this system also revealed that in the model employed here, *PARP1* gene disruption is as toxic as PARPi in a BRCA1-deficient background. It is well

documented that *PARP1* mutations are able to confer resistance to PARPi due to relieving the trapping potential of these agents[7]. Of note, however, these studies were performed in *BRCA1*-proficient backgrounds[5,59,60], or cells harbouring mutations in *BRCA1* that retain residual function[61], perhaps indicating that different genetic

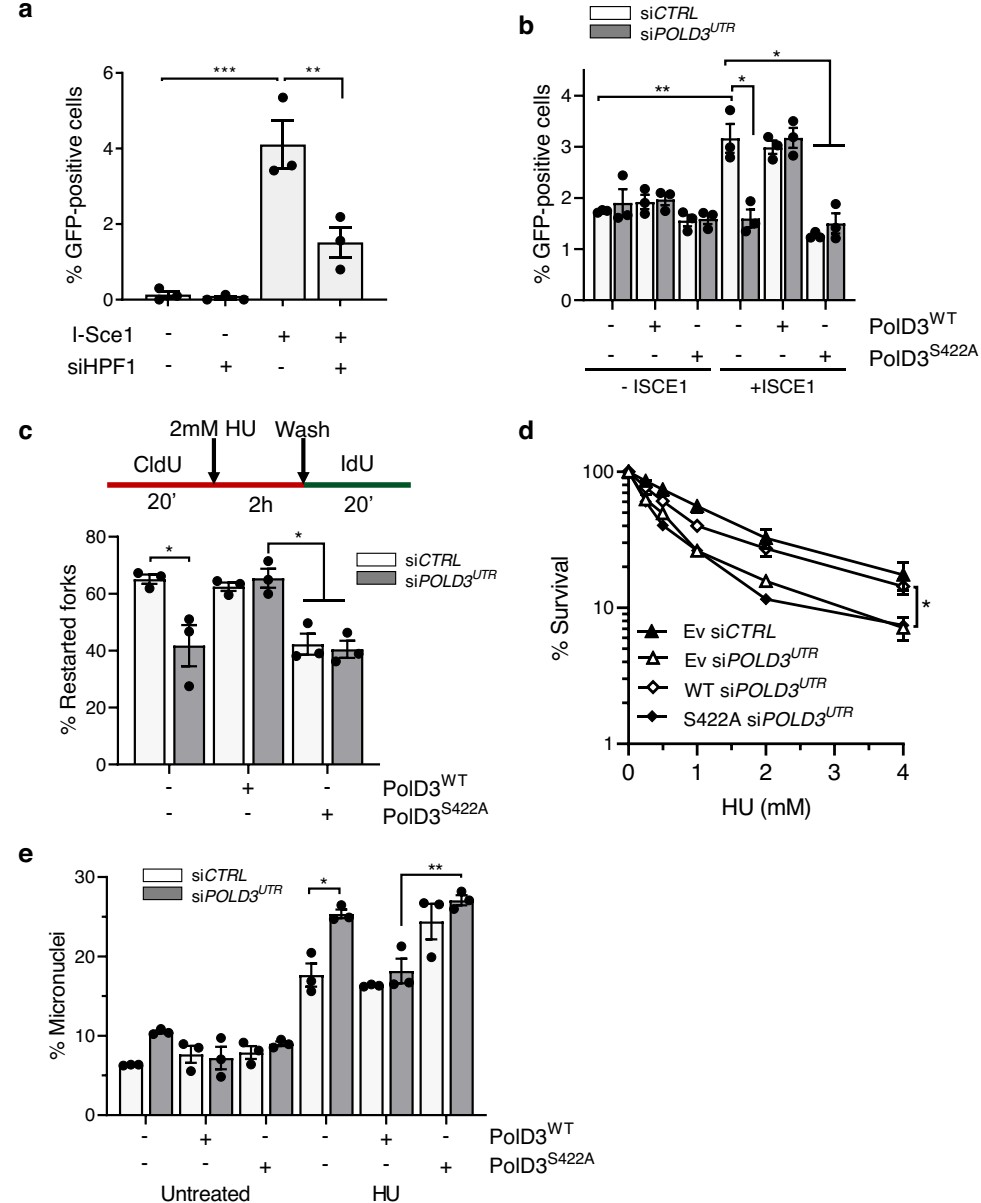

**Fig. 6 | PolD3 S422 is required for efficient BIR and replication fork restart to promote recovery from HU-induced replication stress. a** BIR activity at an I-Sce1-induced break assayed using a GFP-based plasmid reporter stably integrated into U2OS-BIR$_2$ cells, either in the presence or absence of siRNA depletion of HPF1 as indicated. GFP-positive cells were scored by live fluorescence imaging 72 h after transient *I-SCE1* transfection. **b** BIR activity at an I-Sce1-induced break assayed using a GFP-based plasmid reporter stably integrated into U2OS cells expressing siRNA-resistant PolD3$^{WT}$ or mutant PolD3$^{S422A}$, either in the presence or absence of siRNA depletion of endogenous PolD3 as indicated. Where no recombinant PolD3 is indicated, cells were transfected with empty vector. GFP-positive cells were scored by live fluorescence imaging 72 h after transient *I-SCE1* transfection. **c** DNA fibre analysis showing the frequency of stalled forks which restart after release from 2 h HU (2 mM) in FLAG-HA-PolD3 expressing cell lines. Where no recombinant PolD3 is indicated, cells were transfected with empty vector. At least 250 structures were scored in each experiment. **d** Clonogenic survival assay assessing HU sensitivity of cells transfected with empty vector (Ev), or expressing siRNA-resistant PolD3$^{WT}$ or mutant PolD3$^{S422A}$ following siRNA depletion of endogenous PolD3 as indicated. **e** Quantification of micronuclei formation in cells stably expressing siRNA-resistant PolD3$^{WT}$ and mutant PolD3$^{S422A}$ following recovery from 24 h HU exposure either in the absence or presence if siRNA-depletion of PolD3 as indicated. Where no recombinant PolD3 is indicated, cells were transfected with empty vector. At least 400 cells were scored in each experiment. Statistical significance denotes differences in survival following 4 mM HU exposure. In all instances error bars represent the S.E.M of three independent experiments. Statistical analysis performed using a two-tailed Student's *t*-test (**d**) or one-way ANOVA with Tukey's test applied post-hoc (**a**, **b**, **c** and **e**). Statistical significance was calculated to a confidence of 95%, $p < 0.05$ (*), 99%, $p < 0.01$ (**) or 99.9%, $p < 0.001$ (***). Otherwise, analyses were classified as not significant (ns). Source data are provided as a Source Data file.

interactions occur in cells harbouring null mutations, such as those described here.

Our data uncover a novel genetic interaction between PARP1 and Rad52 in supressing the toxicity of HR defects. Similar to the role of Rad52 in yeast, vertebrate Rad52 can act redundantly with BRCA2 to load Rad51 at DNA lesions[28,62], providing one explanation for the synthetic lethal interaction between Rad52 and BRCA2. However, the RPA and Rad51 binding domains of Rad52 are not required for its DNA strand exchange activity, or to maintain cell viability in BRCA-deficient backgrounds[30,31]. Additionally, the synthetic lethal interaction of Rad52 extends to canonical HR factors in backgrounds that are *BRCA2*-proficient[29,32,33]. Together, these data suggest additional regulatory mechanisms that contribute towards the synthetic lethal interaction with HR. A major function for Rad52 in vertebrates is to regulate BIR, a

homology-directed repair mechanism that is required for ALT[36,37] and replication-associated repair either during S-phase[13,24–27] or mitosis[38,39]. Whilst PARG activity and ADP-ribose metabolism regulate ALT, in contrast to our observations excessive ADPr disrupts homology-directed repair in this context[63]. Further, *parp1/2Δ* cells are able to perform MiDAS, indicating that ADPr is also dispensable for BIR during mitosis (Supplementary Fig. 5). Instead, our data point to a critical role for PARP1/2 in regulating BIR and Rad52-dependent replication fork repair (Figs. 2, 3). Given PARPi induce replication stress in *BRCA*-deficient cells, it is tempting to speculate that one aspect of PARPi toxicity is mediated through disruption of BIR-dependent replication fork recovery mechanisms.

In addition to functioning in the same pathway with regards to synthetic lethality with HR, our data clearly point to a role for PARP1/2 in regulating RAD52-dependent replication-repair to suppress DNA damage and genome instability. Restart of collapsed forks by BIR occurs downstream of Mre11-dependent fork degradation and requires Mus81-directed cleavage of forks prior to PolD3-dependent DNA synthesis[13,25,26,64,65]. Rad52 is implicated at several stages of this pathway including preventing excessive remodelling of stalled replication forks, coordination of Mus81 cleavage events, and catalysing strand invasion[24,40,66,67]. It also suppresses alt-NHEJ repair of replication-associated DNA breaks until the onset of mitosis[68]. Here we provide evidence that Rad52 is engaged at stalled/damaged replication forks to promote BIR and that this is facilitated by PARP1/2-dependent ADPr. Moreover, our data indicate that whilst PARP-activation, and as a consequence assembly of Rad52 at stalled/damaged replication forks is independent of Mus81, it requires Mre11 and ATM activity (Fig. 3 and Supplementary Fig. 7b). The disruption of ADPr through the Mre11 inhibitor Mirin indicates Mre11 nuclease activity is required for PARP1/2 activation, suggesting processing of replication forks contribute to PARP1/2 activation. However, given Mre11-dependent ATM activity is required to remove potentially toxic NHEJ factors from one-ended DSBs[42–44], it may also indicate that remodelling of repair factors at the stalled/damaged fork contribute to ADPr events and assembly of BIR proteins at one-ended DSBs.

Given replication fork regression serves as an entry point for Mre11 processing of replication forks[13,17], our data indicating Mre11 is required for HU-induced ADPr demonstrate a role for PARP1/2 in regulating BIR at later time points following replication fork processing. Intriguingly, PARP1 and Rad52 have also been implicated at earlier time points in replication fork recovery by stabilising regressed forks[12,66], suggesting temporally distinct roles for these factors in replication fork recovery. In this regard, mono-ADPr constitutes a second wave of signalling following Poly-ADPr in response to IR[69] and we similarly observe rapid activation of PARPs in response to HU that then plateaus until stronger ADPr at later time points (Fig. 4b). Together, these data suggest distinct roles for ADPr at different times during the remodelling and/or repair of replication forks. Additionally, whilst Rad52-dependent BIR has been studied in the context of excessive replication stress caused by oncogenic activation, or replication fork instability in HR-defective backgrounds[13,24–26,40], we observe a requirement for PARP1/2 in Rad52-dependent replication fork recovery in HR-proficient backgrounds. This indicates a requirement for these pathways more broadly during normal cell cycle progression either at different stages in replication fork remodelling/repair, or in response to different stresses such as blocked/regressed forks or replication-associated DSBs.

Our proteomics data uncover a further level of BIR regulation through PolD3 ADPr. PolD3 is most consistently ADP-ribosylated at S458 in cells exposed to $H_2O_2$[49–52]. In contrast, our data indicate PolD3 is mono-ADPr at S422 in response to HU and that mutation of this site results in loss of ADPr, defective BIR, replication fork recovery, HU sensitivity and genome instability (Figs. 5 and 6). Intriguingly, we observe that PolD3 is also mono-ADPr in the absence of HU and that

mutating S422 has little impact on this modification (Fig. 5a). This indicates that PolD3 is mono-ADPr on a site(s) other than S422 in the absence of replication stress, but upon HU exposure this is lost, either through removal by ARH3[58] or conversion to Poly-ADPr, and S422 is modified. Whether the removal of PolD3 mono-ADP has any functional significance in BIR and/or other repair mechanisms remains to be tested. Nevertheless, what our data indicate is that in addition to promoting assembly of Rad52 at DNA lesions, PARP1/2 regulate replication fork recovery by ADP-ribosylating PolD3 at S422 to promote DNA synthesis during BIR. Additionally, these data suggest distinct ADPr events being induced upon different stresses that regulate functionally distinct outcomes.

In summary, we identify ADPr regulates Rad52-dependent replication fork recovery by BIR. We also uncover that Rad52 and PARP1 function in the same pathway to promote cell viability in the absence of HR. We provide a mechanistic understanding of this process by identifying that Mre11-dependent processing of stalled replication forks is required for HU-induced ADPr that promotes BIR not only by assembly of Rad52 at replication forks, but also through PARP1/2-dependent ADPr of PolD3. Together, these data provide fundamental insights into how cells process stalled and/or damaged replication forks and how these pathways integrate to promote DNA damage tolerance that will help refine strategies to target DDR pathways in the clinic.

## Methods
### Materials
Materials requests should be made to the corresponding author Nick Lakin (nicholas.lakin@bioch.ox.ac.uk).

### Cell lines, cell culture, and genome editing
U2OS cells were cultured in DMEM supplemented with 10% foetal bovine serum at 37 °C in a humid atmosphere containing 5% $CO_2$. The generation of U2OS *parp1Δ*, *parp2Δ* and *parp1/2Δ* cells has been described previously[23]. U2OS *BRCA1*^SMASh cells were generated through CRISPR-Cas9-based insertion of a SMASh degron sequence on the endogenous *BRCA1* gene for the expression of a C-terminal fusion protein product. A CRISPR gRNA (CTAGGGGGTGTCGGTGATG) targeting the site of insertion designed using the CRISPR design tool (http://crispor.tefor.net/), cloned into pSpCas9(BB)−2A-Puro (PX459) V2.0 and transfected into U2OS cells with a homology template containing the SMASh degron sequence flanked by 500 bp arms of sequence homologous to the insertion site. Subsequent *PARP1* disruption in these cells was performed as described previously[23] to generate *BRCA1*^SMASh*parp1Δ* clones (*BRCA1*^SMASh*parp1Δ*.B and *BRCA1*^SMASh*parp1Δ*.C). A sequence verified expression construct containing cDNA sequence for *POLD3* (NCBI RefSeq: NM_006591.2, Sino Biological, #HG19572-UT) was targeted for Q5® Site-directed mutagenesis kit as per the manufacturer's instructions (NEB, #E0552S). A pair of non-overlapping primers introduced a base substitution necessary for PolD3 Ser422Ala mutation. cDNA-containing sequences were cloned into a pLENTI-EF1α-N-FLAG-HA-PURO destination vector (a gift of Ross Chapman) (Gateway Recombination Cloning technology, Invitrogen). To generate stably expressing U2OS cell lines, empty vector, *POLD3*^WT and *POLD3*^S422A expression constructs were co-transfected with third generation packaging constructs in HEK293T cells for viral production. U2OS cells were transduced with harvested lentiviral supernatants and placed under antibiotic selection for 12 days.

### siRNA transfections
Transfections were performed using Dharmafect-1 (Dharmacon) and with 50 nM total siRNA. *BRCA2, DNA2*, and *POLD3* were depleted using an ON-TARGETplus SMARTpool™ (Dharmacon) containing four different siRNA sequences. Individual siGENOME siRNA was used to

deplete *MUS81*. *MRE11* was targeted using a custom siRNA sequence (5′-GAACCUGGUCCCAGAGGAGdTdT-3′). Stably expressing lentivirus cells were transfected with individual ON-TARGETplus siRNAs (Dharmacon) against the 3′-untranslated region of *POLD3*. The relevant non-targeting siRNAs were used as negative controls. Cells were transfected once, again after 24 h, and left to recover for 24 h before seeding for downstream applications. To monitor BIR downstream of siRNA-mediated depletions, cells were seeded 10 h after the 2nd transfection hit.

## BIR GFP plasmid reporter assay

A *pBIR-GFP* plasmid reporter, a gift of Thanos Halazonetis, (addgene # 49807) was stably integrated into U2OS, *parp1/2Δ* and stably expressing PolD3 ADPr mutant cell lines. Independent U2OS-*BIR-GFP* clones (U2OS-*BIR₁*, U2OS-*BIR₂*) were selected from the U2OS-*BIR-GFP* polyclonal population after seeding at low density. Cells were transfected with an *I-SCE1* expression construct, *pI-SCE1-GR-RFP* (a gift from Tom Misteli, addgene #17654). After 8 h, cells were re-seeded into 6 well format. Next day, cells were treated with 1 μM triamcinolone acetonide (TA) to induce I-Sce1 nuclear localisation and, where appropriate, PARP inhibitors. After 48 to 72 h, GFP-expressing cells were analysed by flow cytometry or imaged by microscopy and analysed using ImageJ.

## Immunofluorescence

Cells were seeded on to glass coverslips to attach overnight. Where applicable, cells were pulsed with 1 μM for 1 h prior to DNA damage. After HU treatments, cells were pre-extracted in CSK buffer (10 mM HEPES, pH 7.8, 300 mM sucrose, 3 mM $MgCl_2$, 0.5% Triton) for 4 min at 22 °C or 0.5% Triton/PBS for 5 min at 4 °C before fixation in 4% PFA for 20 min at 4 °C. For MiDAS experiments, cells were treated with aphidicolin (Cayman) and ATR inhibitor VE-821 (Sigma) in the presence of the CDK1 inhibitor RO3306 (Sigma) to promote G2/M arrest overnight. Cells were released into mitosis in the presence of 10 μM EdU where appropriate. Mitotic cells were simultaneously permeabilised and fixed by incubation in PTEMF (20 mM PIPES, pH 6.8 10 mM EGTA 0.2% Triton, 1 mM $MgCl_2$, 4% formaldehyde) for 20 min at 22 °C. For the detection of 53BP1 bodies in G1, cells were fixed in paraformaldehyde without pre-extraction. Fixed cells were permeabilised in 0.5% Triton/PBS for 5 min at 4 °C before blocking. Except for PAN-ADP-ribose detection, which required blocking with 2% FBS/PBS-T, all coverslips were incubated in 3% BSA/PBS for 1 h. Where applicable, incorporated EdU was conjugated to fluorescent azide dye by incubation in a click reaction buffer (100 mM Tris-Cl, pH 8.0, 4 mM $CuSO_4$, 50 mM sodium ascorbate, 20 μM azide dye) for 30–120 min depending on the experiment. Immunostaining and cell imaging was performed as before[23] using a Zeiss IX70 and 10x or 100x oil immersion objective lens. To score anaphase defects in mitosis and micronuclei formation in interphase, cells were exposed to HU where appropriate, fixed in PFA, permeabilised and mounted in DAPI-containing medium. All structures were scored manually at the microscope. Antibodies are listed in Supplementary Data 2.

## Western blotting and biochemical fractionation

For whole cell extracts, cells were harvested by trypsinisation, washed twice in ice-cold PBS, pelleted by centrifugation, and boiled in boiling in 1x SDS loading buffer. Chromatin extracts were prepared as described previously[31]. All protein extracts were resolved by SDS-PAGE, transferred on to PVDF membranes (Millipore, 0.45 μM pores) and blocked with 5% milk/TBS-T. Membranes were probed with primary antibodies overnight, washed extensively and incubated with relevant HRP-conjugated secondary antibodies for 1 h (IRDye 800 CW-or IRDye 700 CW-Secondary (1Li-cor)). After further washing, luminescence signal generated using Immobilon Western Chemiluminescent HRP substrate (Millipore) was detected using autoradiography or the Odyssey® XF Imaging system (LI-COR Biosciences). LI-COR images

were quantified with Image Studio™. Antibodies are listed in Supplementary Data 2.

## Assessing ADP-ribosylation status of proteins by affinity purification or immunoprecipitation

Stably expressing PolD3^WT and PolD3^S422A cells were treated with 2 mM HU for 24 h. After treatment, cells were washed twice in ice-cold PBS, collected by scraping and washed once more. Following centrifugation (600 g, 5 min), cell pellets were lysed in Modified RIPA Buffer (50 mM Tris-Cl, pH 7.5, 300 mM NaCl, 1% NP-40, 0.5% sodium deoxycholate, 1 mM EDTA) supplemented with 75 μM tannic acid (Sigma), 2 mM β-glycerophosphate, 5 mM NaF, 2 mM $Na_3VO_4$, 40 μM PJ-34 (Selleck) and 10 μM PARG inhibitor PDD 00017273 (Tocris). Clarified lysate was diluted 1:2 (vol/vol) in modified RIPA without salt but supplemented with additives. ADP-ribose pulldown was performed by incubating lysates with Protein G DynaBeads™ (Invitrogen) pre-bound to 20 μg PAN-ADP-ribose binding reagent (Merck, MABE1016). After 4 h at 4 °C, beads were washed in modified RIPA buffer without detergents (4 °C, 3 × 10 min) before elution by gentle agitation with equal volume 2 x SDS loading buffer for 20 min at 22 °C. ADP-ribosylated enriched proteins were analysed by SDS-PAGE.

In the case of the FLAG-POLD3 immunoprecipitation, anti-FLAG M2 magnetic beads were incubated with U2OS whole cell extracts (lysed in 200 mM NaCl-modified RIPA buffer, see above) for 2 hr at 4 °C and washed 3 times in lysis buffer, before the elution steps. Inputs (5%) and FLAG-precipitates were blotted by SDS-PAGE using mono-ADPr (Biorad AbD33205ad) and POLD3 antibodies. Antibody-free beads were used as a negative control.

## Clonogenic survival assays

Cells were counted with a haemocytometer before plating 6-well plates and left overnight. The next day cells were treated with HU or ASV. After 24 h, HU was removed, cells were extensively washed in PBS, then left to recover in fresh media for 10–14 days. For ASV treatments, cells were constantly exposed for 10–14 days. Media was refreshed every 2–3 days. After incubation, cells were fixed in 100% methanol for 20 min at −20 °C and stained with 0.5% crystal violet (Sigma) for 20 min at 22 °C. Colonies of >50 cells were scored as viable. Colony survival was calculated as a percentage relative to untreated conditions.

## Pulse-field gel electrophoresis

To prepare agarose plugs, approximately $5 \times 10^6$ cells were cultured per condition. After HU treatment, cells were harvested by trypsinisation, washed in ice-cold PBS, and pelleted by centrifugation. Resuspended cells ($8 \times 10^6$ cells/ml in 10 mM Tris-Cl, pH 7.2, 20 mM NaCl, 50 mM EDTA) were equilibrated to 50 °C before mixing 1:1 (vol/vol) with a pre-warmed 2% low melting agarose solution. Cell/agarose mixtures were quickly but gently homogenised by pipetting, then transferred to CHEF disposable plug moulds (Bio-Rad, #1703713). Each sample produced 2–3 plugs embedded with $8 \times 10^5$ cells. Solidified plugs were digested in Proteinase K Reaction Buffer (100 mM EDTA, pH 8.0, 0.2% sodium deoxycholate, 1% sodium lauryl sarcosine, 1 mg/ml Proteinase K) without agitation for 24 h at 50 °C. After extensive washing (20 mM Tris-Cl, pH 8.0, 50 mM EDTA), DNA in sample plugs and *S. cerevisiae* standards (Bio-Rad, #1703605) were resolved in 0.7% agarose gels in 0.5xTBE. Electrophoresis was performed for 21 h at 14 °C using a CHEF DR III Variable Angle System (Bio-Rad) set to the following parameters: Block 1: 9 h, 120° included angle, 5.5 V/cm 30–18 s switch; Block II: 6 h, 117° included angle, 4.5 V/cm, 18–9 s switch; Block III: 6 h, 112° included angle, 4.0 V/cm, 9–5 s switch. DNA was visualised by staining gels with ethidium bromide (1 μg/ml). Gels were destained in distilled water overnight before visualisation using a GelDoc Imaging System (Bio-Rad).

## DNA fibre analysis

Cells were pulsed with 25 μM CldU for 20 min, treated with 2 mM HU for 2 h, extensively washed in warm media, before second labelling with 250 μM IdU for 20 min. Cells were washed twice in ice-cold PBS and harvested by trypsinisation. To prepare extended DNA fibres, pelleted cells were resuspended in PBS to a final concentration of $5 \times 10^5$ cells/ml. Next, 2 μl cell suspension were spotted on to microscope slides, air-dried for 5 min, and lysed with 7 μl DNA Spreading buffer (200 mM Tris-Cl, pH 7.4, 50 mM EDTA, 0.5% SDS). After gentle mixing, lysates were incubated for 2 min before slides were tilted by 15° to allow spreading. Air-dried slides were fixed in methanol/acetic acid (3:1) for 15 min. DNA fibre spreads were immunolabelled as described elsewhere (Ronson et al., 2018). Fibres were examined using a Zeiss IX70 microscope with 100x oil-immersion objective lens. Images were processed and analysed in ImageJ (FIJI). For each experimental sample, at least 250 fibres were classified.

## Mass spectrometry analysis of ADP-ribosylated peptides

$2.5 \times 10^8$ U2OS cells were left untreated or exposed to 2 mM HU. Cells were washed twice in ice-cold PBS, harvested by gentle scraping, and collected by centrifugation (600 g, 4 °C, 5 min). Pellets were suspended in 10 pellet volumes of lysis buffer (6 M Gnd-HCl, 50 mM Tris-HCl, pH 8.5) with alternating cycles of disruption by shaking and vortexing for 30 s. Samples were processed as described previously[52]. Briefly, lysates were homogenised by sonication then pre-digested using Lys-C followed by trypsin digestion overnight prior to peptide purification and lyophilisation. Potential PAR chains were reduced through PARG incubation overnight before affinity purification using a GST-tagged Af1521 macrodomain. TFA-eluted peptides were Stage-Tipped prior to MS analysis. In brief, all samples were analysed using an 80 min gradient on a nanoscale EASY-nLC 1200 system connected to an Orbitrap Fusion Lumos mass spectrometer (ThermoFisher). Full scans were performed at a resolution of 120,000 and an injection time of 250 ms, with MS/MS spectra measured in the Orbitrap at a resolution of 60,000 and an injection time of 1000 ms. All RAW files were processed using the MaxQuant software suite version 1.5.3.30[70] allowing ADPr on C, D, E, H, K, R, S, T, and Y.

## Statistical analysis

The relevant statistical tests are stated where applicable. In each case statistical significance was analysed to 95% (*), 99% (**) and 99.9% (***) confidence intervals. Exact P-values are provided in the Source Data file. Otherwise, analyses were classified as not significant (ns). Student's t-tests were two-tailed and unpaired. For multiple comparisons one-way analysis of variance (ANOVA) was performed with Tukey's test applied post-hoc. Graphs and analysis were performed in Excel 2016.

## Reporting summary

Further information on research design is available in the Nature Portfolio Reporting Summary linked to this article.

## Data availability

All data generated or analysed during this study are included in this published article and its supplementary information files. The mass spectrometry proteomics data have been deposited to the ProteomeXchange Consortium via the PRIDE partner repository with the dataset identifier PXD035661. Source data are provided with this paper.

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

## Acknowledgements

We thank Ivan Ahel (University of Oxford) for the *HPF1* and *ARH3* knockout cells and Thanos Halazonetis for the BIR reporter constructs. This

research was funded in whole, or in part, by the Wellcome Trust (Grant 102348/Z/13/Z) and MRC (Grants MR/P028284/1; MR/P018963/1; MR/V00896X/1; MR/W017350/1). For the purpose of Open Access, the author has applied a CC BY public copyright licence to any Author Accepted Manuscript version arising from this submission. JL was supported by AstraZeneca. M.L.N. lab is supported by the Novo Nordisk Foundation (NNF14CC0001; NNF13OC0006477), Danish Council of Independent Research (4002-00051, 4183-00322 A, 8020-00220B; 0135-00096B) and The Danish Cancer Society (R146-A9159-16-S2). The proteomics technology was funded by the EU's Horizon 2020 research and innovation programme (EPIC-XS-823839).

## Author contributions

All experiments were performed by F.R. except BRCA1SMASh cell generation and characterisation (J.L.), MS analysis (S.C.B.-L. and M.L.N.). Analyses using ATMi and HPF1/ARH3 cells and selected BIR assay data were performed by M.J.L.-C. with assistance from N.S.. M.J.L.-C and A.B.S. performed ADP-ribosylation analysis of PolD3. N.D.L., F.R., and S.C.B.-L. wrote the manuscript.

## Competing interests

The authors declare no competing interests.
