## [Peer Review File · Nature Communications]

Regulation of Rad52-dependent replication fork recovery through serine ADP-ribosylation of PolD3Editorial Note: Parts of this Peer Review File have been redacted as indicated to remove third-party material where no permission to publish could be obtained.

REVIEWER COMMENTS

Reviewer #1 (Remarks to the Author):

Richards et al reported the results to suggest that PARP1/PARP2 dependent poly ADP ribosylation (ADPr) depends on the Mre11 activity and promotes stalled replication fork recovery by Rad52-, PolD3-dependent BIR. Overall, the results are compelling and especially the identification of a new ADPr site in PolD3 critical for the recovery of stalled replication fork after HU treatment is exciting; it could help reveal in part how PARP1 regulates fork stability and restart after stalling. Nevertheless, the current report is somehow disjointed as the results are not fully integrated into the testable model and insufficient to provide further details, making it a lot to be desired still. I am also concerned that authors simply assume the primary function of Rad52 in BIR without ample experimental supports and did not make reasonable efforts to test or acknowledge other possibilities.

1. It is puzzling why PARP1 deletion and olaparib sensitivity is equally synthetic to BRCA1 depletion even though it is well documented that olaparib sensitivity in BRCA deficient cells is even partially rescued by PARP1 deletion. In fact, in page 8, line 264, the authors indicated that "under certain conditions" PARP1 disruption is as toxic as PARPi in a BRCA1 deficient background. What are these conditions and why under these conditions, the effect of PARP1 disruption is no less toxic than PARP1 trapping? It requires some discussions.
2. It will be helpful to show the level of BRCA1 depletion when various doses of ASV were treated in Fig. 1d.
3. I suggest to include more detailed descriptions of BIR assay in the text and to include controls to validate the assay outcomes.
4. Does PARPi inhibit both PARP1 and PARP2? It is curious if the disruption of both PARP1/2 causes stronger synthetic interaction with BRCA1 shutoff as the deletion of *parp1/parp2* in fig 2 leads to more severe defects in Fig.2 and 3.
5. Is BIR more defective in *rad52*^{-/-} *parp1*^{-/-} or *rad52*^{-/-} cells after *parpi* treatment?
6. In Fig, 2C, is the broken DNA detection by gel electrophoresis saturated?
7. Does the effect of Mre11 depletion on the damage induced ADPr depend on ATM? This needs to be addressed.
8. Is *mre11* inhibition synthetic to BRCA1 depletion?
9. Because PolD3 is likely required for general HR as well, the author should test if *pol3-S422A* is defective in HR by DR-GFP assay.
10. Does PolD3 ADPr depend on Mre11 and/or Mus81?

Reviewer #2 (Remarks to the Author):

The authors present an interesting connection between PARP activity, Break induced replication, and poly(ADP) ribosylation of PolD3.

The best evidence for a direct connection to BIR involves quantification of this process using a substrate, but there are critical flaws with this data. The authors do not describe this substrate

sufficient for a reader to understand it (a citation isn't enough). They do not validate the structure of GFP+ products (e.g. using southern analysis, or PCR). The representation of data in Figure 2a doesn't make sense; why isn't this represented like Figure 2b? There is no direct connection made between Mre11 depletion and PARP-dependent BIR; this needs to be performed directly, measuring BIR using the same criteria for other experiments (the substrate), since differences in Rad52 foci observed in Figure 3 may reflect other roles for Rad52.

In Figure 5b and 5c, the authors fail to discuss the surprising defect observed in siCTRL treated, PolD3 S422A expressing cells. Why are the control treated cells defective?

There are problems with statistical analyses. When studying three or more groups, multiple t-tests of cherry-picked pairs is associated with Type I error. Please use ANOVA with correction for multiple hypothesis testing, for example. This is particularly problematic for Figure 2d and 2f, where effects are quantitatively modest, relative to the extent of data dispersion.

There are also a number of minor concerns.

Related to the following statement:

"Importantly, this site is either absent⁴⁷, 49, 50, 53 or at low abundance⁴⁶ in studies that employ other genotoxins..." Please specify which genotoxins were used in this cited work.

The following sentence is difficult to understand as written; consider breaking it up.

"However, in support of PARP1/2 regulating a pathway that functions with Rad52 to promote replication fork repair, whilst depletion of Rad52 elevates levels of HU-induced DSBs and nuclear γH2AX, it does not induce a further increase in these phenotypes in *parp1/2Δ* cells (Fig. 2c, d)."

Reviewer #3 (Remarks to the Author):

In this manuscript, Richards et al report a novel role for PARP1/2 in regulating Rad52 dependent fork repair crucial for maintaining cell viability in BRCA1 deficient background. They present data that Rad52 and PARP1/2 are epistatic. Further, they show that Mre11 is the activator of PARPs, which in turn promote Rad52 assembly and break-induced replication (BIR). They map replication stress induced ADP-ribosylation sites and show that PolD3 activity is required for BIR and that PolD3 with its identified ADP-ribosylation site (Ser422) changed to alanine is not functional. The manuscript is well written, and the topic is timely and uncovers novel facets of PARP functions in the regulation of replication stress response, in particular, BIR. The quality of data appears good. The BRCA Smash approach is very elegant. This should be of interest for the readership of Nature Communications. Yet, the study in its present form falls short of elucidating the regulation of Rad52-dependent replication fork recovery through serine ADP-ribosylation of PolD3, which the title promises.

Specific remarks

The data on figures 1-4 are very good and their interpretation sound. It would have been great to see more raw data presented at least as supplementary figures. In most cases, the reviewer needs to rely on bar plots.

However, based on the data on figure 5, it is not possible to link PARP activity with PolD3 ADP-ribosylation and that the ADP-ribosylation of PolD3 on Ser422 regulates BIR. In Panel A there is data on PolD3 being ADP-ribosylated following HU treatment, this data should be supplemented with HU untreated samples. It would be great to elucidate if PolD3 chromatin loading precedes Rad52 foci formation, is parallel to it or follows it. It would be important to complement data in panel C, using PARPi to reveal that the loading of PolD3 is or isn't regulated by PARP activation, since mutant PolD3 appears less chromatin bound upon HU than WT. It is not clear if the S422A mutation would retain any function of PolD3. It is not shown that the S422A mutant cannot be ADP-ribosylated. Data in panels B, D and F may require additional experiments or good explanation: how is it that the siCTRL

treated cells supplemented with S422A mutant PolD3 are defective, while in those cases wild type endogenous PolD3 should be present – in all three experimental setups, therefore, they are very unlikely to be some sort of labeling mistake? Is the mutant some sort of dominant negative? If yes, its mode of action should be better understood.

In its present form the manuscript does not show that PolD3 is ADP-ribosylated following HU treatment. Maybe the mass-spec analysis but this I did not fully understand – is ADP-ribosylated PolD3 identified in all HU treated conditions or not? It is not shown if S422A abolished PolD3 ADP-ribosylation. It is not known if the ADP-ribosylation of PolD3 on S422 regulates its function or not – only data that show that the S422A mutant is not active. How is BIR and PolD3 activity, and its chromatin binding in ARH3 KOs where PolD3 should be essentially constitutively ADP-ribosylated? What is the phenotype in HPF1 KOs where serine ADP-ribosylation should be essentially absent?

Reviewer #1

Richards et al reported the results to suggest that PARP1/PARP2 dependent poly ADP ribosylation (ADPr) depends on the Mre11 activity and promotes stalled replication fork recovery by Rad52-, PolD3-dependent BIR. Overall, the results are compelling and especially the identification of a new ADPr site in PolD3 critical for the recovery of stalled replication fork after HU treatment is exciting; it could help reveal in part how PARP1 regulates fork stability and restart after stalling. Nevertheless, the current report is somehow disjointed as the results are not fully integrated into the testable model and insufficient to provide further details, making it a lot to be desired still. I am also concerned that authors simply assume the primary function of Rad52 in BIR without ample experimental supports and did not make reasonable efforts to test or acknowledge other possibilities.

We thank this reviewer for their positive comments and that they feel the results are compelling and exciting.

We were disappointed that they found the report disjointed and that it lacked an integrated approach to test our model. Our original submission proposed that PARP1 and Rad52 function in the same pathway to maintain cell viability when homologous recombination is dysfunctional and that this relationship extends to replication fork recovery through BIR. Mechanistically, we identified that Mre11-dependent activation of PARP1/2 regulate Rad52 assembly at stalled/damaged replication forks and that ADP-ribosylation of PolD3 is required for BIR, replication fork recovery and genome stability. To address the concerns of this referee, we now include a significant amount of new data to further validate this model and improve the narrative. Specifically, we provide further controls and validation of the assay to quantify BIR efficiency in a variety of genetic backgrounds. This includes epistasis analysis of PARP1/2 with known BIR proteins to place them in the same DNA repair pathway. As requested, we also provide data that further reinforce the relationship between Mre11 and PARP1/2 in BIR and their synthetic lethal interaction with BRCA1. Finally, we provide new data that confirms the link between ADP-ribosylation and PolD3 at S422 and its requirement for BIR, replication fork recovery and genome stability.

We would like to thank this reviewer for raising these points, which we specifically address below, and hope they agree that this greatly improves the narrative and conclusions of this study.

1. It is puzzling why PARP1 deletion and olaparib sensitivity is equally synthetic to BRCA1 depletion even though it is well documented that olaparib sensitivity in BRCA deficient cells

is even partially rescued by PARP1 deletion. In fact, in page 8, line 264, the authors indicated that "under certain conditions" PARP1 disruption is as toxic as PARPi in a BRCA1 deficient background. What are these conditions and why under these conditions, the effect of PARP1 disruption is no less toxic than PARP1 trapping? It requires some discussions.

With hindsight prompted by this reviewer's comment we realise that our statement 'PARP1 gene disruption is as toxic as PARPi' requires further discussion. We are unclear why we observe this phenotype using our experimental system. Presumably, the differences in phenotypes are due to the different genetic backgrounds of the models used in each study. In this regard, it is striking that all the reports of PARP1 mutations that confer resistance to PARPi are performed either in BRCA-proficient backgrounds, or cells with *BRCA1* mutations that retain residual function. This is in contrast to our cells that were specifically designed to ablate PARP1 and BRCA1 function in combination. We have amended the results (p4) and discussion (p9) to raise this possibility.

2. It will be helpful to show the level of BRCA1 depletion when various doses of ASV were treated in Fig. 1d.

This data has now been included in Supplementary Figure 1b and described in the text (p4).

3. I suggest to include more detailed descriptions of BIR assay in the text and to include controls to validate the assay outcomes.

We have now included a more detailed description of the BIR reporter assay previously developed and validated in the Halazonetis laboratory (Costantino *et al. Science* 343:88), in addition to further controls to validate assay outcomes (see p4/5). Specifically, similar to the original validation of this assay, sequencing of GFP products induced upon I-Sce-1 expression indicate accurate recombination-based DNA repair (see response to Reviewer #2, point 1). Importantly, I-Sce-I-dependent GFP induction is dependent on Rad52 and PolD3 (Figure 2c and 6b, p5), two well established evolutionary conserved BIR genes. Moreover, we show that the reduced GFP induction observed following PARP and Rad52 inhibition is not additive when inhibitors are applied in combination (Figure 2c, p5), indicating these genes function in the same pathway to reconstitute GFP through BIR.

4. Does PARPi inhibit both PARP1 and PARP2? It is curious if the disruption of both PARP1/2 causes stronger synthetic interaction with BRCA1 shutoff as the deletion of

parp1/parp2 in fig 2 leads to more severe defects in Fig.2 and 3.

It is well established that PARP inhibitors target both PARP1 and PARP2. However, we were unclear how to address this comment as we have not directly compared PARP disruption using PARP inhibitors with *parp1/parp2* gene disruption in this manuscript. Whilst we employ PARP inhibitors in the BRCA1^{SMASh} cells, we have only compared this with *parp1* gene disruption in this background (Figure 1d and e). Where we use *parp1/2Δ* knockout cells in Figure 2 and 3 referred to by this reviewer, we have not directly compared these phenotypes with PARP inhibitors. We therefore do not feel in a position to comment on the relative contributions of PARP1/2 disruption through inhibition vs. *parp1/parp2* gene deletion in these data sets.

5. Is BIR more defective in rad52-/- parp1-/- or rad52-/- cells after parpi treatment?

To address this point we have used Rad52 and PARP inhibitors as opposed to gene deletion. As indicated in point 3, PARP and Rad52 inhibitors are not additive for a BIR defect when applied in combination (Figure 2c, p5), indicating these genes function in the same pathway in the BIR assay.

6. In Fig, 2C, is the broken DNA detection by gel electrophoresis saturated?

A longer exposure of the gel is illustrated in the main figure to highlight the emergence of broken DNA following HU exposure (now Figure 2d). However, the quantification of these data was performed on a standard exposure of the gels. A representative image (in addition to the quantification from 3 independent experiments) is now illustrated in the source data file to highlight the broken DNA band being quantified is not saturated.

7. Does the effect of Mre11 depletion on the damage induced ADPr depend on ATM? This needs to be addressed.

ADPr in response to HU is dependent on ATM. These data have been included in the results (Supplementary Figure 5b, p6) and referred to in the discussion (p10/11).

8. Is mre11 inhibition synthetic to BRCA1 depletion?

Yes. These data have now been included in Figure 3i and commented on in the text (p6).

Please also note that in response to Reviewer #2 we have also further validated the link between Mre11 and PARP1/2 in replication fork recovery by BIR. We provide additional data demonstrating that depletion of Mre11 similarly compromises BIR in the GFP reporter assay (Figure 3j, p6). Importantly, within this experiment we also demonstrate that whilst PARP1/2 also compromise BIR, they do not result in a more severe phenotype when administered in combination with Mre11 depletion. This indicates they are epistatic, providing a direct connection between Mre11 depletion and PARP-dependent BIR.

9. Because PolD3 is likely required for general HR as well, the author should test if pol3-S422A is defective in HR by DR-GFP assay.

We did not perform these experiments in the original manuscript as it is already established that PolD3 is not required for general HR, as judged using the DR-GFP assay (Tumini et al., *Scientific Reports* 6:38873 | DOI: 10.1038/srep38873), and assays that specifically measure synthesis-dependent strand annealing or single strand annealing (Costantino et al., *Science* 343:88).

10. Does PolD3 ADPr depend on Mre11 and/or Mus81?

Our data indicating that knockdown of Mre11 (but not Mus81) reduces HU-induced ADPr to levels observed in untreated cells (Figure 3c) and that this results in defective BIR (Figure 3j) suggest PolD3 ADPr is dependent on Mre11. This, taken together with the comments of Reviewer #3 led us to focus our attention on generating significant new data that directly assess factors which influence PolD3 Ser-ADPr (PARP1/2, HPF1, ARH3, PolD3-S422; Figure 5a-d). These data clearly indicate that Ser-ADPr at S422 of PolD3 is required for replication fork recovery by BIR. Therefore, given the time constraints for these revisions we hope this reviewer will understand why we do not include an analysis of whether PolD3 ADPr is dependent on Mre11 in the revised manuscript.

Reviewer #2

The authors present an interesting connection between PARP activity, Break induced replication, and poly(ADP) ribosylation of PolD3.

We thank this reviewer for considering our manuscript represents an interesting connection between PARP activity, BIR and PolD3 ADP-ribosylation. The main points raised by this reviewer concerned the validation and interpretation of the BIR assay. We now include

additional data to address these concerns, in addition to all their other points that we outline below:

1. The best evidence for a direct connection to BIR involves quantification of this process using a substrate, but there are critical flaws with this data. The authors do not describe this substrate sufficient for a reader to understand it (a citation isn't enough). They do not validate the structure of GFP+ products (e.g. using southern analysis, or PCR). The representation of data in Figure 2a doesn't make sense; why isn't this represented like Figure 2b? There is no direct connection made between Mre11 depletion and PARP-dependent BIR; this needs to be performed directly, measuring BIR using the same criteria for other experiments (the substrate), since differences in Rad52 foci observed in Figure 3 may reflect other roles for Rad52.

We now provide further description and controls to further validate the BIR assay, including data that provides a direct link between Mre11 and PARP-dependent BIR. Several points were raised in this paragraph, so for clarity we now deal with each individually below:

The authors do not describe this [GFP reporter] substrate sufficient for a reader to understand it (a citation isn't enough).

We have now included a more detailed description of the BIR reporter assay previously developed and validated in the Halazonetis laboratory (Costantino *et al. Science* 343:88; see p4/5).

They do not validate the structure of GFP+ products (e.g. using southern analysis, or PCR).

We now include additional data to validate the BIR assay that we describe in p4/5. As requested, we have amplified and sequenced the GFP cassette reconstituted in the reporter following I-Sce-1 expression. Consistent with previous validation of the assay (Costantino *et al. Science* 343:88), this confirms reconstitution of the GFP gene across the I-Sce1 site, indicating accurate recombination-based DNA repair (Reviewer Figure 1). The assay is specifically designed to exclude repair by synthesis-dependent strand-annealing or single-strand annealing, making repair and reconstitution of the GFP cassette BIR-dependent. To confirm this, in our original submission we demonstrated I-Sce1-dependent GFP reconstitution is dependent on the BIR factor PolD3 (Figure 5b in the original manuscript – now Figure 6b). We further validate this assay by demonstrating that this event is also dependent on another critical BIR protein, Rad52 (Figure 2c). Importantly, we also show that the reduced GFP induction observed following PARP and Rad52 inhibition is not additive

when inhibitors are applied in combination (Figure 2c), indicating these genes function in the same pathway to reconstitute GFP through BIR.

[redacted]

Reviewer Figure1: Reconstitution of GFP in the BIR assay

a. Schematic representation of the inverted *GFP* break-induced replication assay reporter cassettes originally described and validated by Costantino *et al.* (*Science* **343**, 88-91; 2014). The 3 segments on the GFP gene that recombine through shared homology in segment 2 are indicated, as is the I-Sce1 restriction enzyme site. PCR primers that amplify reconstituted GFP are indicated. **b.** Sequence alignment of the GFP gene (GFP; upper sequence) with GFP the PCR product amplified from cells following expression of I-Sce1 (BIR_GFP; lower sequence) indicating the segments recombine into their correct order.

The representation of data in Figure 2a doesn't make sense; why isn't this represented like Figure 2b?

The data in Figure 2a use U2OS and *parp1/2Δ* cells that needed to be generated in parallel. They are therefore a distinct set of cells from the USOS-BIR clones generated for use in the other assays that employ either inhibitors or siRNA. The cells employed in Figure 2a have a higher background of GFP-positive cells than the U2OS-BIR clones. Therefore, for clarity we expressed these data as relative GFP induction, as opposed to % GFP-positive cells.

There is no direct connection made between Mre11 depletion and PARP-dependent BIR; this needs to be performed directly, measuring BIR using the same criteria for other experiments (the substrate), since differences in Rad52 foci observed in Figure 3 may reflect other roles for Rad52.

As requested, we provide additional data demonstrating that depletion of Mre11 similarly compromises BIR in the GFP reporter assay (Figure 3j, p6). Importantly, within this

experiment we also demonstrate that whilst PARPi also compromise BIR, they do not result in a more severe phenotype when administered in combination with Mre11 depletion. This indicates they are epistatic, providing a direct connection between Mre11 depletion and PARP-dependent BIR.

2. In Figure 5b and 5c, the authors fail to discuss the surprising defect observed in siCTRL treated, PolD3 S422A expressing cells. Why are the control treated cells defective?

The data outlined in the original Figure 5 (now Figure 6 in the revised manuscript) show that siRNA depletion of PolD3 causes defects in BIR and replication fork restart that is reflected in sensitivity to HU and elevated micronuclei. In this PolD3-depleted background the phenotypes are clearly and reproducibly rescued by expression of wild-type PolD3, but not the S422A mutant. It is therefore clear that PolD3-S422A is unable to support replication fork recovery through a BIR-based mechanism, the key finding in this figure. This also raises the possibility that PolD3-S422A acts as a dominant negative mutation in a wild-type PolD3 background. Importantly, we include additional data demonstrating that exogenous PolD3-S422A interacts with other components of the PolD complex (PolD1 and PolD2) as efficiently as wild-type PolD3 (Supplementary Figure 9b). This supports the possibility that the defects observed in siCTRL conditions are due to a dominant negative effect of PolD3-S422A. This possibility has been raised in the revised manuscript (p9).

3. There are problems with statistical analyses. When studying three or more groups, multiple t-tests of cherry-picked pairs is associated with Type I error. Please use ANOVA with correction for multiple hypothesis testing, for example. This is particularly problematic for Figure 2d and 2f, where effects are quantitatively modest, relative to the extent of data dispersion.

Where appropriate, these analyses have been performed and stated in the figure legend. This does not change the interpretation of the data. An exception is the original Figure 2d, where despite there being an increase in γ H2AX upon depletion of Rad52 that is significant in our original analysis, it loses significance using the recommended analysis suggested by this referee. It is therefore not possible to perform epistasis analysis on these data, so we no longer felt comfortable using this data in the manuscript. Importantly, however, we stress that our more direct analysis of DNA breaks formed in response to replication stress (revised Figure 2d) clearly shows a significant increase in DNA breaks upon Rad52 depletion, or in *parp1/2* Δ cells, and that this is epistatic. Therefore, our overall conclusion that Rad52 and

PARP1/2 suppress replication associated DNA damage and genome instability through a shared mechanism remains unchanged.

There are also a number of minor concerns:

Related to the following statement: “Importantly, this site is either absent^{47, 49, 50, 53} or at low abundance⁴⁶ in studies that employ other genotoxins...” Please specify which genotoxins were used in this cited work.

All these studies utilised H₂O₂. We have modified the text to reflect this (p8).

The following sentence is difficult to understand as written; consider breaking it up. “However, in support of PARP1/2 regulating a pathway that functions with Rad52 to promote replication fork repair, whilst depletion of Rad52 elevates levels of HU-induced DSBs and nuclear γ H2AX, it does not induce a further increase in these phenotypes in *parp1/2* Δ cells (Fig. 2c, d).”

We have modified this sentence as requested (p5).

Reviewer #3

In this manuscript, Richards et al report a novel role for PARP1/2 in regulating Rad52 dependent fork repair crucial for maintaining cell viability in BRCA1 deficient background. They present data that Rad52 and PARP1/2 are epistatic. Further, they show that Mre11 is the activator of PARPs, which in turn promote Rad52 assembly and break-induced replication (BIR). They map replication stress induced ADP-ribosylation sites and show that PolD3 activity is required for BIR and that PolD3 with its identified ADP-ribosylation site (Ser422) changed to alanine is not functional. The manuscript is well written, and the topic is timely and uncovers novel facets of PARP functions in the regulation of replication stress response, in particular, BIR. The quality of data appears good. The BRCA Smash approach is very elegant. This should be of interest for the readership of Nature Communications. Yet, the study in its present form falls short of elucidating the regulation of Rad52-dependent replication fork recovery through serine ADP-ribosylation of PolD3, which the title promises.

We thank this reviewer for their constructive comments and considering that the manuscript is well-written, timely, and uncovers novel facets of PARP functions in the regulation of replication stress response. We now include significant new data to address their comments, as described below. Importantly, with regards their comment that the manuscript falls short

of elucidating the regulation of replication fork recovery through PolD3 ADP-ribosylation, we now include the assessment of PolD3 function in HPF1 and ARH3 cells, in addition to a robust analysis of the ADP-ribosylation status of mutant PolD3 (PolD3^{S422A}) in response to hydroxyurea (HU).

Specific remarks

1. The data on figures 1-4 are very good and their interpretation sound. It would have been great to see more raw data presented at least as supplementary figures. In most cases, the reviewer needs to rely on bar plots.

We have included all raw data in the Source data file.

2. However, based on the data on figure 5, it is not possible to link PARP activity with PolD3 ADP-ribosylation and that the ADP-ribosylation of PolD3 on Ser422 regulates BIR. In Panel A there is data on PolD3 being ADP-ribosylated following HU treatment, this data should be supplemented with HU untreated samples. It would be great to elucidate if PolD3 chromatin loading precedes Rad52 foci formation, is parallel to it or follows it. It would be important to complement data in panel C, using PARPi to reveal that the loading of PolD3 is or isn't regulated by PARP activation, since mutant PolD3 appears less chromatin bound upon HU than WT. It is not clear if the S422A mutation would retain any function of PolD3. It is not shown that the S422A mutant cannot be ADP-ribosylated. Data in panels B, D and F may require additional experiments or good explanation: how is it that the siCTRL treated cells supplemented with S422A mutant PolD3 are defective, while in those cases wild type endogenous PolD3 should be present – in all three experimental setups, therefore, they are very unlikely to be some sort of labeling mistake? Is the mutant some sort of dominant negative? If yes, its mode of action should be better understood.

In response to these comments, in addition to those raised in point 2 below, we now provide new data that link PARP activity, PolD3 ADP-ribosylation on Ser422 and regulation of BIR.

Several different points were raised with regards to Figure 5 in this point. For clarity we deal with each of these individually below:

In Panel A there is data on PolD3 being ADP-ribosylated following HU treatment, this data should be supplemented with HU untreated samples.

We now provide data assessing the ADP-ribosylation status of PolD3 +/- HU - see response to point 2 below.

It would be great to elucidate if PolD3 chromatin loading precedes Rad52 foci formation, is parallel to it or follows it.

To address this comment we have included data to assess the kinetics of PolD3 enrichment into chromatin, as requested (Supplementary Figure 10). This indicates assembly of PolD3 into chromatin begins at similar time points post administration of HU as the detection of Rad52 nuclear foci – 4-8 hours and 6-12 hours respectively. However, these data should be treated with caution as the resolution of the assays are not sufficient to detect subtle differences in the kinetics of loading profiles. It should also be stressed that this comparison employs different assays/reagents that will result in different detection thresholds. Therefore, whilst the two approaches assess the same phenomenon (i.e. assembly of Rad52/PolD3 into chromatin-bound structures in response to HU), they are not directly comparable. As such, we do not feel comfortable commenting on whether PolD3 chromatin loading precedes Rad52 foci formation, is parallel to, or follows it. We have therefore been conservative with our interpretation of these data and limit our comment in the manuscript to ‘we observe enrichment of PolD3 in chromatin following exposure of cells to HU with similar kinetics to that observed for Rad52 foci formation’ (p8).

It would be important to complement data in panel C, using PARPi to reveal that the loading of PolD3 is or isn't regulated by PARP activation, since mutant PolD3 appears less chromatin bound upon HU than WT.

We now include data demonstrating assembly of PolD3 in chromatin is sensitive to PARP inhibition (Figure 5b and p8).

It is not clear if the S422A mutation would retain any function of PolD3. It is not shown that the S422A mutant cannot be ADP-ribosylated.

We now provide data indicating that the polD3 S422A mutant is not mono-ADPr following treatment with HU - see response to point 2.

Data in panels B, D and F may require additional experiments or good explanation: how is it that the siCTRL treated cells supplemented with S422A mutant PolD3 are defective, while in those cases wild type endogenous PolD3 should be present – in all three experimental setups, therefore, they are very unlikely to be some sort of labeling mistake? Is the mutant some sort of dominant negative? If yes, its mode of action should be better understood.

As suggested by this reviewer, we believe the defects observed in siRNA control cells expressing PolD3-S422A is due to a dominant negative affect. We include additional data demonstrating that exogenous PolD3-S422A interacts with other components of the PolD

complex (PolD1 and PolD2) as efficiently as wild-type PolD3 (Supplementary Figure 9b). This supports the possibility that the defects this reviewer refers to in siCTRL conditions are due to a PolD3-S422A competing with wild-type PolD3 to induce a dominant negative phenotype. This possibility has been raised in the revised manuscript (p9).

2. In its present form the manuscript does not show that PolD3 is ADP-ribosylated following HU treatment. Maybe the mass-spec analysis but this I did not fully understand – is ADP-ribosylated PolD3 identified in all HU treated conditions or not? It is not shown if S422A abolished PolD3 ADP-ribosylation. It is not known if the ADP-ribosylation of PolD3 on S422 regulates its function or not – only data that show that the S422A mutant is not active. How is BIR and PolD3 activity, and its chromatin binding in ARH3 KOs where PolD3 should be essentially constitutively ADP-ribosylated? What is the phenotype in HPF1 KOs where serine ADP-ribosylation should be essentially absent?

This reviewer is right to highlight the importance of independently assessing the ADPr status of wild-type and S422A PolD3 either in the absence or presence of HU. As requested, we have now performed these experiments by immunoprecipitation of wild-type or PolD3^{S422A} from untreated cells, or cells exposed to HU. Recombinant PolD3 serine mono-ADPr levels were then assessed by western blotting with an antibody that detects HPF1-dependent mono-ADPr (Figure 5a, p8). These data clearly illustrate that whilst wild-type PolD3 is mono-ADPr with or without HU, mono-ADPr of PolD3^{S422A} is absent following induction of replication stress. Together, these data indicate that whilst S422 is not a major ADPr acceptor in unstressed cells, it plays a significant role in determining the mono-ADPr status of PolD3 in response to HU.

With regards whether ADP-ribosylation of PolD3 on S422 regulates its function or not, we now provide additional data to address this point. As requested, we include data demonstrating that HU-induced assembly of PolD3 in chromatin is sensitive to PARP inhibitors and disruption of the HPF1 gene (Figure 5b, c), indicating a requirement for PARP catalytic activity and Ser-ADPr in assembly of PolD3 at sites of stalled/damaged replication forks. This notion is reinforced by our observations that cells disrupted in ARH3, the gene required for removal of Ser-ADP, exhibit elevated levels of PolD3 in chromatin fractions (Figure 5d). Further, we included data that illustrate BIR activity is dependent of HPF1 (Figure 6a). Together, these data support a role for Ser-ADPr in PolD3-dependent BIR and are presented in Figure 5b-d, Figure 6a and p8/9.

REVIEWER COMMENTS

Reviewer #1 (Remarks to the Author):

The revision addresses points raised by the reviewers, and thus significantly improved. Identification of damage induced ADPr and the targets will be important contributions to the field, justifying its values to the scientific community. Only two fold increase in BIR assay results poses some concerns on discriminating the effect of gene deletion or inhibitor treatment on conditions indirect to the assay outcomes (i.e. cell cycle, gene expression) from their true roles in BIR. It is also confusing how the current results can be integrated to fork reversal and/or TMEJ, both are known to involve the factors described in this manuscript.

Reviewer #2 (Remarks to the Author):

Major concerns.

1) Upon closer examination, it doesn't appear that any of the siRNA experiments have been validated for depletion: this critical omission needs to be corrected for all such experiments.

2) Regarding Fig. 2A- especially given that the pair of lines are isogenic, there is no reason to alter the method of data representation. The authors must report the unprocessed data (fraction of cells that are GFP-positive); the comparison isn't valid otherwise.

3) New information relating to PolD3 ribosylation is difficult to resolve. The authors argue S422 is a major site of ribosylation in response to replication stress. They then show PolD3 is mono ADP ribosylated constitutively, both in wild type and in S422A mutant PolD3, and missing from S422A POLD3 only after HU treatment (what happened to the constitutive ribosylation at other sites?). It isn't clear why Poly(ADP) ribosylation wasn't assessed. Most importantly, the functional significance of S422 as a ribosylation target remains questionable. The authors should be able to demonstrate that cells expressing S422A are refractory to the effects of PARPi on BIR.

Minor Concerns.

4) It remains difficult for readers that haven't previously worked in BIR to understand how this substrate functions. While the added text included is better than nothing, please include a diagram (supplemental if desired), showing the initial substrate, intermediates, and expected products.

5) Line 255: "Importantly, this site is either absent...."
This is a mis-statement; the site isn't absent – ribosylation at that site is.

6) Some of the differences observed in chromatin loading (e.g. Fig 5D) are modest. The authors need to show evidence of a reproducible and significant difference, much as they did in Fig 5e.

Reviewer #3 (Remarks to the Author):

The resubmitted manuscript has answered my questions and should be published.

There is one minor question regarding Figure 5d: in the chromatin fraction of the ARH3 KO cells there is ARH3 specific (is it?) band revealed by the anti-ARH3 antibody. please check that.

Reviewer #1

The revision addresses points raised by the reviewers, and thus significantly improved. Identification of damage induced ADPr and the targets will be important contributions to the field, justifying its values to the scientific community. Only two fold increase in BIR assay results poses some concerns on discriminating the effect of gene deletion or inhibitor treatment on conditions indirect to the assay outcomes (i.e. cell cycle, gene expression) from their true roles in BIR. It is also confusing how the current results can be integrated to fork reversal and/or TMEJ, both are known to involve the factors described in this manuscript.

We thank this reviewer for considering the manuscript is significantly improved.

With regards to the comment on two-fold increases in our BIR assay making data difficult to interpret, we stress that we generally see an induction of at least 5-6 fold in the majority of these assays (Figures 2b&c, 3j, 6a). Where this is not the case (Figures 2a and 6b), the differences we highlight are statistically significant. We therefore feel our conclusions are valid.

This reviewer is right to highlight that some of the factors analysed here (PARP1, Rad52) also regulate TMEJ and replication fork reversal. Given the focus of this study is concerned with regulation of BIR by PARP1/2, we are reluctant to speculate too much about how these other pathways integrate with the processes investigated here. Having said this, we realise the importance of discussing these points, particularly with respect to replication fork reversal/recovery, and have modified the discussion to consider these possibilities (p10-11).

Reviewer #2

Major concerns.

1) Upon closer examination, it doesn't appear that any of the siRNA experiments have been validated for depletion: this critical omission needs to be corrected for all such experiments.

We now include representative images for knockdown efficiency using each of the siRNAs employed in this study (Supplementary Figure 3).

2) Regarding Fig. 2A- especially given that the pair of lines are isogenic, there is no reason to alter the method of data representation. The authors must report the unprocessed data (fraction of cells that are GFP-positive); the comparison isn't valid otherwise.

The difficulty with this experiment is that integration of the GFP reporter cassette results in different levels of background GFP signal in the two different cell lines (see Figure 2a source data for U2OS and *parp1/2Δ* cells without I-Sce1 induction). This is not a problem with experiments that employ inhibitors or siRNA in a single cell line such as the U2OS-BIR clones (e.g. Figures 2b and c, 3j, 6a and b). However, when comparing the U2OS cells with *parp1/2Δ* cells it is difficult to make a direct comparison unless this background level of GFP is taken into account. It is for this reason that we expressed the induction of GFP in each cell line upon expression of I-Sce1. This is a readout of how a given cell line can repair an I-Sce1-induced DSB by BIR - a significant induction of GFP in response to I-Sce1 indicates a cell line is able to perform BIR, whilst no significant induction represents an inability to perform BIR.

Having said this, we understand the importance of consistent representation of data when using the same assay in a study. We still feel the data expressing relative induction of GFP in these cells is important, but have shifted this to Supplementary Figure 4. Instead, we now include the representation of % GFP cells in Figure 2a, as requested. In this analysis the only significant difference is between the +/- I-Sce1 data sets in U2OS cells. Therefore, whilst these cells are able to perform BIR, the *parp1/2Δ* cell line is not. The text has been adjusted accordingly to reflect these revisions (p5).

3) *New information relating to PolD3 ribosylation is difficult to resolve. The authors argue S422 is a major site of ribosylation in response to replication stress. They then show PolD3 is mono ADP ribosylated constitutively, both in wild type and in S422A mutant PolD3, and missing from S422A POLD3 only after HU treatment (what happened to the constitutive ribosylation at other sites?). It isn't clear why Poly(ADP) ribosylation wasn't assessed. Most importantly, the functional significance of S422 as a ribosylation target remains questionable. The authors should be able to demonstrate that cells expressing S422A are refractory to the effects of PARPi on BIR.*

We assessed mono-ADPr (as opposed to poly-ADPr) as this is the predominant ADP-ribosylation event put onto serines (Leidecker *et al. Nature Chemical Biology* **12**, 998; Bonfiglio JJ, *et al., Molecular cell* **65**, 932; Palazzo *et al., Elife* **7**, e34334). We did attempt assessing Poly-ADPr but the reagents that detect this modification result in too many non-specific bands in this analysis to make the data interpretable.

With regards to the comment concerning the constitutive ADPr of PolD3, in our view the critical observation in Figure 5a is that in contrast to wild-type PolD3, mono-ADPr of the

S422A mutant is almost entirely absent when cells are exposed to HU. We therefore believe our statement that S422 is the major site of PolD3 mono-ADPr in response to replication stress is valid. What our data also indicate is that PolD3 is mono-ADPr on a site(s) other than S422 in the absence of replication stress, but upon HU exposure this is lost and instead S422 is modified. The reviewer raises an interesting point regarding what happens to the PolD3 mono-ADPr site(s) present in untreated cells once HU is administered. Presumably, this is removed by ARH3, the enzyme that de-ribosylates serine (Fontana et al., *Elife* **6**, e28533), or converted to Poly-ADPr. However, whilst interesting, we feel that mapping this site(s), distinguishing the two possible removal routes, and establishing the functional significance of this event and how it relates to S422 ADPr warrants a study in its own right and is beyond the scope of the current manuscript. Therefore, we hope this reviewer understands that we have limited our revisions to raising these important points in the discussion (p11-12).

We agree that the experiment to assess whether cells expressing S422A are refractory to PARPi is another mechanism to support a link between S422 ADPr and regulation of BIR. Unfortunately, our attempts at this experiment have been unsuccessful. It should be noted that these cells have a multitude of detrimental manipulations including over-expression of wild-type or dominant negative PolD3^{S422A}, I-Sce-1 over-expression and siRNA knockdown that makes a further insult through exposure to PARPi intractable. We highlight that we illustrate the S422A mutation renders PolD3 refractory to ADPr in response to HU and that this interferes with a variety of its functions including BIR and replication fork restart, in addition to cell viability and genome stability in response to HU. Given these observations provide a direct link between PolD3 ADPr status, BIR, replication fork recovery and genome stability, in addition to the technical difficulties associated with the experiment requested, we hope this reviewer understands that we have not included the suggested analysis in the revised manuscript.

Minor Concerns.

4) *It remains difficult for readers that haven't previously worked in BIR to understand how this substrate functions. While the added text included is better than nothing, please include a diagram (supplemental if desired), showing the initial substrate, intermediates, and expected products.*

We clearly reference the original manuscript in which this assay was developed (Costantino L, et al., *Science* **343**, 88-91). This report has a clear diagram showing the original design of the reporter cassette, in addition to the predicted intermediates and outcomes of BIR

induced by I-Sce-1 cleavage. As such, if we were to include a diagram of the assay it would essentially be a reproduction of this figure. Whilst we are happy to include a more detailed description of the assay, due to potential issues of plagiarism and out of respect to the authors of the original study we are not comfortable reproducing a previously published figure. We therefore feel it is more appropriate to cite the original paper in which the assay was developed and validated, allowing the reader to access this information if required.

5) Line 255: “Importantly, this site is either absent...”

This is a mis-statement; the site isn't absent – ribosylation at that site is.

We have modified this statement to ‘Importantly, ADPr of this site is either absent.....’ (p8)

6) Some of the differences observed in chromatin loading (e.g. Fig 5D) are modest. The authors need to show evidence of a reproducible and significant difference, much as they did in Fig 5e.#

We provide a repeat of the experiment illustrated in Figure 5d and quantification of the data in Supplementary Figure 13 and refer to this in the text (p8).

Reviewer #3

The resubmitted manuscript has answered my questions and should be published.

There is one minor question regarding Figure 5d: in the chromatin fraction of the ARH3 KO cells there is ARH3 specific (is it?) band revealed by the anti-ARH3 antibody. please check that.

We thank this reviewer for recommending the manuscript for publication.

With regards to the comment concerning the ARH3 band in Figure 5d, we see a clear band at the predicted molecular weight for ARH3 (39 kDa) in cytoplasmic fractions of wild-type but not ARH3 knockout cell lines. This band runs slightly higher than the two bands apparent in chromatin fractions, which are non-specific. In our experience ARH3 is at low levels in chromatin and therefore difficult to detect. For clarity we have labelled the ARH3 band in the revised figure.